# Ultra-high field fMRI identifies an action-observation network in the common marmoset

Alessandro Zanini [1✉], Audrey Dureux [1], Janahan Selvanayagam[2] & Stefan Everling [1,2]

The observation of others' actions activates a network of temporal, parietal and premotor/prefrontal areas in macaque monkeys and humans. This action-observation network (AON) has been shown to play important roles in social action monitoring, learning by imitation, and social cognition in both species. It is unclear whether a similar network exists in New-World primates, which separated from Old-Word primates ~35 million years ago. Here we used ultra-high field fMRI at 9.4 T in awake common marmosets (*Callithrix jacchus*) while they watched videos depicting goal-directed (grasping food) or non-goal-directed actions. The observation of goal-directed actions activates a temporo-parieto-frontal network, including areas 6 and 45 in premotor/prefrontal cortices, areas PGa-IPa, FST and TE in occipito-temporal region and areas V6A, MIP, LIP and PG in the occipito-parietal cortex. These results show overlap with the humans and macaques' AON, demonstrating the existence of an evolutionarily conserved network that likely predates the separation of Old and New-World primates.

---

[1] Centre for Functional and Metabolic Mapping, Robarts Research Institute, University of Western Ontario, London, ON, Canada. [2] Department of Physiology and Pharmacology, University of Western Ontario, London, ON, Canada. ✉email: azanini4@uwo.ca

Understanding the intentions and behavior of other individuals is crucial for our daily social interactions. When we observe other individuals carrying out a certain action, we are often able to draw conclusions about their intentions and/ or the purpose of their actions. Functional imaging studies have shown that action observation activates a large-scale network of brain areas known as the action-observation network (AON)[1–3]. In Old-World primates, the core of the AON includes three main nodes: the occipito-temporal region corresponding to the superior temporal sulcus (STS), the inferior parietal lobule (IPL or PFG/AIP) and a prefrontal/premotor cluster (human's ventral premotor cortex and inferior frontal gyrus, macaque's F5 and area 45)[1,4–16].

The AON has been described as a hierarchical system (but with both feedforward and feedback connections)[17] which contains areas activated by progressively greater complexity[18]. Visual neurons in the STS are sensitive to action observation processes and integrate multiple low-features, such as shape and motion[5,7,19–22]. This represents the visual input of the AON which is subsequently transmitted to the rest of the network through projections to visual and visuomotor neurons in the IPL[7,9,23–25]. The parietal and premotor nodes of the AON represent observed actions according to their goal[5–7,9,12,14,25–29], enabling understanding of others' intentions through a process of social action monitoring and facilitating the planning of an adequate behavioral response.

The three nodes described above represent the classic AON core, but more recent studies suggest that this network may be more extensive[12], including clusters in dorsal premotor and motor cortices, somatosensory areas (SI and SII), ventrolateral prefrontal cortex and in the superior parietal lobule (IPS), in both humans[1,15,30–32] and macaques[33–38].

Support for a putative AON in New World monkeys has come from an electrophysiological study by Suzuki and colleagues, who reported a small number of neurons in the ventrolateral frontal region and in the STS of the common marmoset (*Callithrix jacchus*) that were activated by action observation[39]. Marmosets live in family groups and shares some interesting similarities with humans, including prosocial behavior, imitation, and cooperative breeding. Their small size combined with recent wireless and datalogger recording techniques[40,41] make it possible to record high density neural activity during natural social behavior in this species. Their lissencephalic (smooth) cortex offers the opportunity for laminar electrophysiological recordings and optical imaging in many cortical areas[42]. These features could make marmosets an excellent primate model for the study of the neural basis of action observation.

To identify a putative AON in marmosets, we employed whole-brain ultra-high-field fMRI at 9.4 T in awake marmosets. Similar to previous macaque studies[6,7], we investigated the activations induced by the presentation of videos representing the upper limb of a marmoset performing a goal-directed (reaching and grasping for food) or non-goal-directed arm and hand movement (reaching and grasping to an empty space).

## Results

In this fMRI study, we used a block-design in which each run contained 8 video sequences belonging to four different experimental conditions, alternating with baseline blocks. The videos belonging to the Grasping Hand condition showed the upper limb of a marmoset performing a complete reaching-to-grasp movement of a small piece of food (i.e., marshmallow). In the Empty Hand condition, a similar movement was shown, but in this case the piece of food was absent, rendering the action non-goal-directed. The other two experimental conditions (Scrambled

Grasping Hand and Scrambled Empty Hand) were the phase-scrambled version of these two classes of videos (see Methods section for details). Functional maps of the contrasts between each condition and baseline are displayed in Supplementary Figure 4. Seven common marmosets were included in the study, using 10 runs per animal in the analyses (70 runs in total). In the eye-tracking experiment controlling for differences in oculomotor patterns, the same Grasping and Empty Hand videos were presented in a sound-attenuating chamber to 10 common marmosets.

**An occipito-temporal cluster to process vision of body-parts in motion.** We first performed an analysis to identify brain activations induced by videos displaying intact movements (i.e, a movement of an arm and hand), without considering the goal of the action, (Grasping and Empty Hand conditions), versus phase-scrambled videos of these movements (Scrambled Grasping and Empty Hand conditions). The results of this group-level analysis are shown in Fig. 1 (paired $t$ test, results reported at $p < 0.001$ and corrected for cluster-size threshold via Monte Carlo simulations, with α=0.05. See Supplementary Note 1 for a description of the individual maps and for each single condition activation map, and refer to Supplementary Figs. 1 and 2). We found that the observation of the intact movement videos elicited stronger activations both in the occipital (in high-level visual areas, such as V3, V4, and V4T, in particular in the left hemisphere) and temporal cortices. In the latter case, we observed two large bilateral activation clusters, including area FST, the PGa-IPa conjunction, and a wide portion of the TE complex (TE1, TE2, TE3, and TEO) in the left hemisphere, as opposed to a less extensive contralateral activation (including almost all of the TE3 area). Significant activations at the level of $p < 0.001$ but not extended enough to resist cluster-wise correction can be observed at the level of the bilateral prefrontal cortex, including areas 45 and 47, and the left premotor cortex (6Va and 6Vb).

These results, depicted in Fig. 1, show regions activated by videos of grasping movements with or without goal-directed actions, consistent with previously described body patches in the marmoset temporal cortex[43,44].

**An action-observation network for goal-directed actions.** Whereas the occipito-temporal node of the AON is mainly involved in processing several low-level features of the action observed, the parietal and premotor/prefrontal nodes process actions in terms of their goal[5,7]. To test whether areas in the marmoset brain show a similar pattern, we compared the activations elicited by the Grasping condition with the Empty Hand condition (paired $t$ test, results reported at $p < 0.001$ with cluster-size threshold determined by Monte Carlo simulations for $α = 0.05$. See Supplementary Fig. 2 for individual maps). The results show a large network of fronto-temporo-parietal regions that is strongly activated by goal-directed actions, compared to a similar but non-goal directed motor act (Fig. 2). The premotor/prefrontal cluster includes rostral and caudal dorsal premotor cortices (6DR and 6DC) and the ventral premotor cortex (6Va), bilaterally. These clusters extend to a small portion of dlPFC (8C in both hemispheres, plus right 8aV) and to the vlPFC (bilateral area 45, plus wide portions of the right area 47). In the right hemisphere, we observed that this cluster also extends to area 13 in the OFC, while in the left hemisphere there is a region more active for goal-directed actions in the more rostral part of the motor cortex (4ab).

A similar bilateral pattern of activation is observed at the occipito-parietal junction, in which stronger responses to goal-directed actions are present in higher-level visual areas (V3A, V6

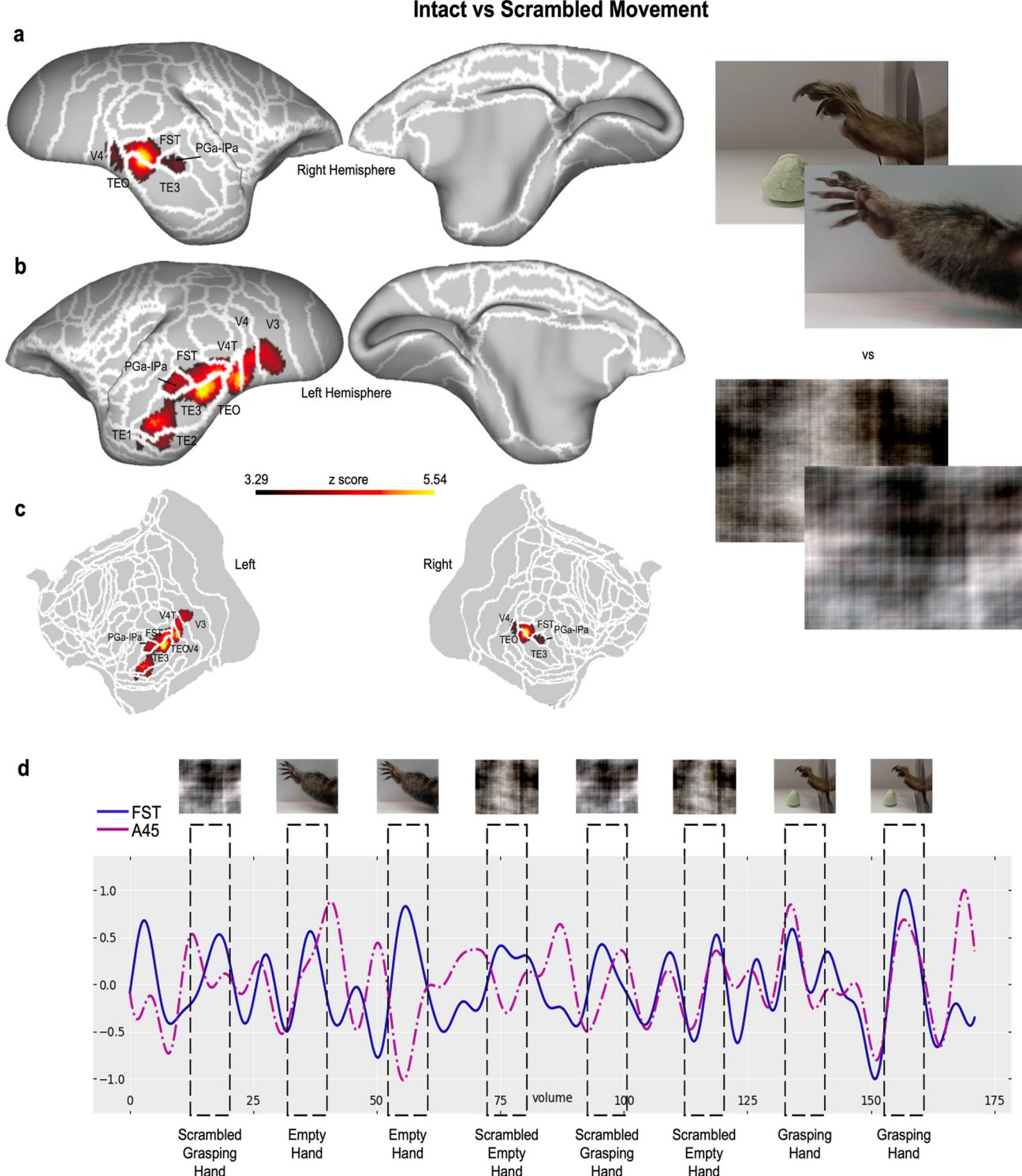

**Fig. 1 Functional maps of the contrasts between the Intact versus the Scrambled Motion conditions.** Maps are displayed on the left (**a**) and right (**b**) fiducial brain surfaces (lateral view on the left, medial view on the right) and on flat maps (**c**, left hemisphere on the left, right hemisphere on the right). White lines delineate the cerebral areas included in the Paxinos parcellation of the NIH marmoset brain atlas[107]. The responses here reported have intensity higher than $z = 3.29$ (corresponding to $p < 0.001$, AFNI's 3dttest + +) and survived the cluster-size correction (10,000 Monte-Carlo simulations, $a = 0.05$). **d** depicts the BOLD timecourse of one exemplary run for two key regions of the marmoset's AON: area 45 (purple dashed line) and FST (blue solid line). The black dashed rectangles show the onset, duration and offset of the experimental blocks.

and V6A) and in parietal areas MIP and LIP, with a greater extension of these latter activations in the left hemisphere. Here, we also find an extension of the cluster to parietal area PG, at the border to LIP.

Finally, the third and largest bilateral activation cluster is found in the occipito-temporal cortex, where stronger responses to an observed goal-directed action are located in both visual (V2, V3, V4, V4T and V5) and in more temporal regions

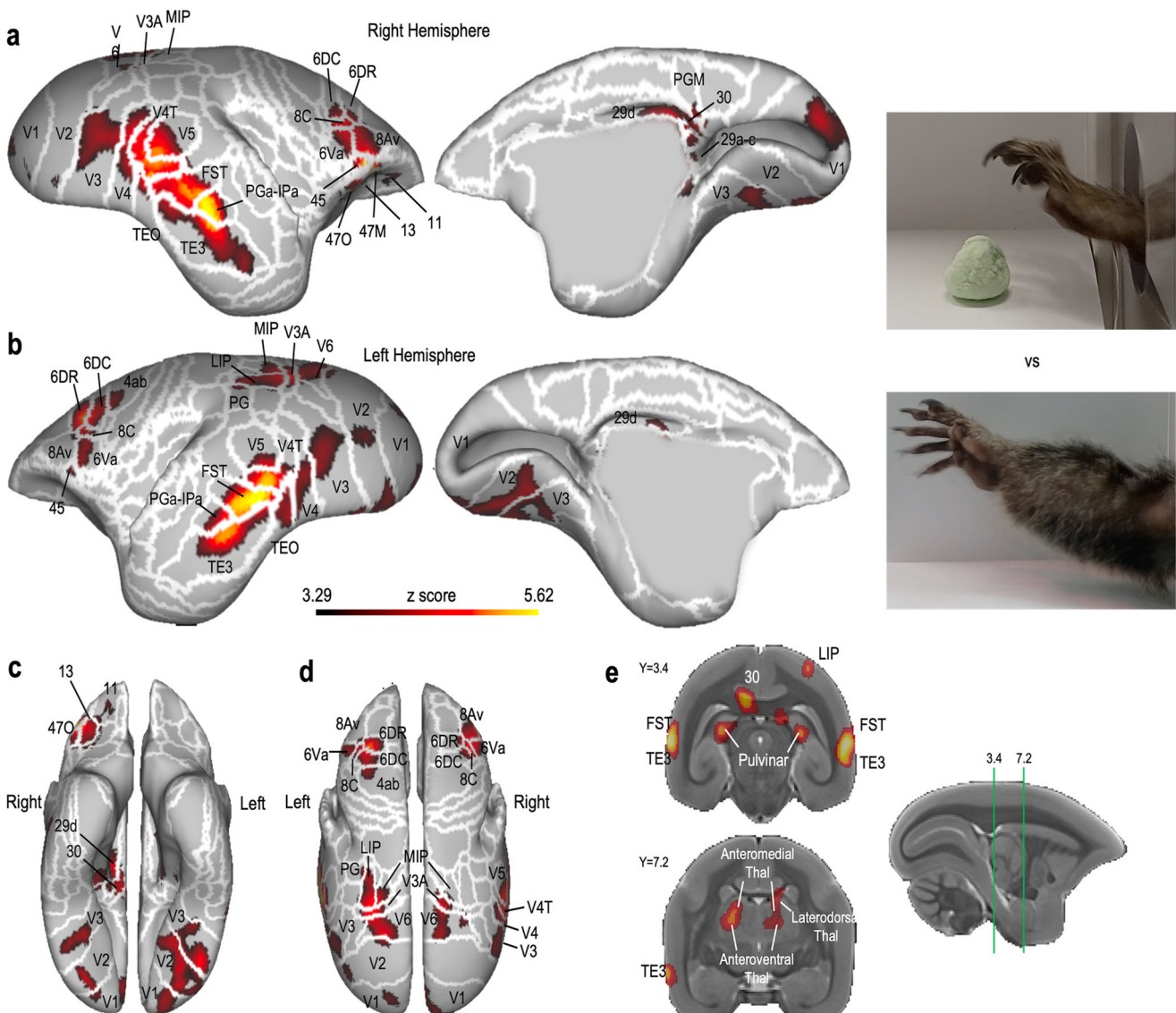

**Fig. 2 Functional maps of the contrasts between Grasping Hand and Empty Hand conditions.** Maps are displayed on the left (**a**) and right (**b**) fiducial brain surfaces (lateral view on the left, medial view on the right, ventral view in **c**, dorsal view in **d**). Particular of the subcortical thalamic activations in **e**. White lines delineate the cerebral areas included in the Paxinos parcellation of the NIH marmoset brain atlas[107]. The responses here reported have intensity higher than $z = 3.29$ (corresponding to $p < 0.001$, AFNI's 3dttest + +) and survived the cluster-size correction (10,000 Monte-Carlo simulations, $a = 0.05$).

including the most dorsal part of TEO, the FST, the PGa-IPa and the TE3.

In addition to these bilateral activations, we also observed a unilateral activation in the right hemisphere, in the posterior cingulate region, corresponding to medial areas 29, 30 and PGM.

Subcortically, we found bilateral activations in the hippocampal formation and in the thalamus. In the right hemisphere, a first cluster included the anteromedial and anteroventral thalamic nuclei, extending into the laterodorsal and ventrolateral dorsal thalamic nucleus. In the left hemisphere, we found activations in the laterodorsal and in ventrolateral thalamic nuclei. Both this clusters extended posteriorly, including wide portions of the bilateral pulvinar.

These results demonstrate the existence of a widespread fronto-temporo-parietal network activated by the observation of goal-directed reaching-to-grasp movements, similar to

the AON already described in both humans and macaque monkeys[1,5–7,12,14,23,45,46].

**Similar eye movement patterns for body parts in motion.** Figure 3 reports the results of the analyses conducted on the frequency and amplitude of saccades during the viewing of the Grasping and Empty hand videos outside of the scanner. An example of eye movement traces for each type of video is available in Supplementary Fig. 5. Two separate Wilcoxon signed rank tests were conducted on these two variables to investigate potential differences in oculomotor patterns during the two main experimental conditions. As shown in Fig. 3, the eye-tracking experiment did not reveal any significant difference in the frequency of the saccades ($W = 26$, $p = 0.92$) nor in saccade amplitude ($W = 23$, $p = 0.70$) between the Grasping and Empty hand videos. Moreover, considering the presence of a target in the

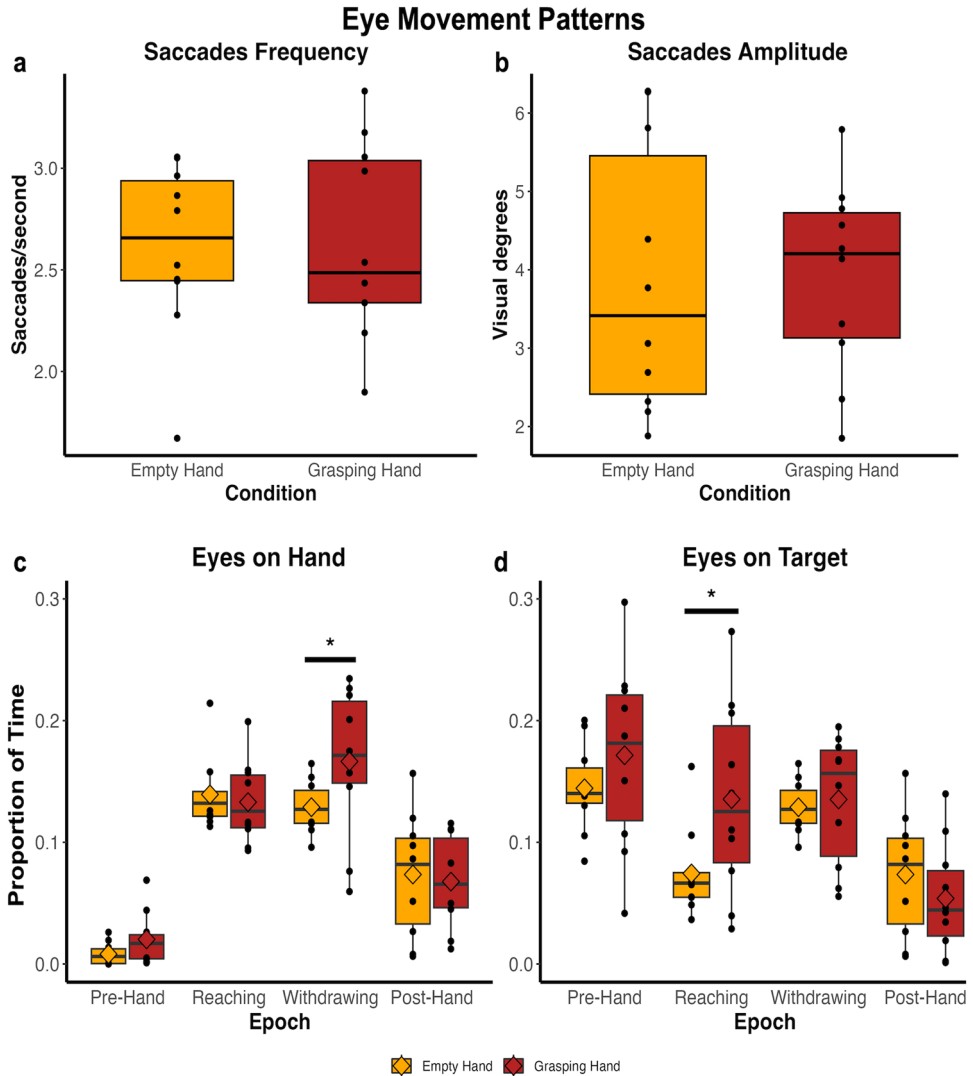

**Fig. 3 Eye movement patterns in goal-directed and non-goal-directed conditions.** Frequency (**a**, number of eye movements per second with a radial eye velocity greater than 30 deg/s and amplitude greater than 0.35 visual degrees) and amplitude (**b**, in visual degrees) of saccades performed during the presentation of videos depicting goal-directed (Grasping Hand condition, in red) and non-goal-directed actions (Empty Hand condition, in orange). No differences between conditions emerged using Wilcoxon signed rank test (in both cases, $p > 0.6$). **c**, **d** represent the proportion of time spent observing a window of 3 visual degrees centered on the hand or target, respectively. This proportion is reported for the two experimental conditions (Empty Hand, in orange, and Grasping Hand, in red) across the four epochs (pre-hand, reaching, withdrawing and post-hand). Post-hoc comparisons with Bonferroni correction showed a significant difference between the two experimental conditions during the withdrawing phase regarding the visual window centered on the hand (**c**) and during reaching for the visual window centered on the target (**d**). Asterisks indicate such significant differences: *$p < 0.05$. Black lines and squares represent median and mean of the group respectively; black dots represent individual scores. Bars on bottom and top of the boxes represent the first and third quartile respectively.

Grasping Hand videos and its absence in the Empty Hand condition, we investigated separately the eye movements towards the limb and target positions throughout the duration of the videos (the target position in the Empty Hand condition was considered as the position of the maximum arm extension achieved in each video, see more in details in Methods section). According to our hypothesis, the presence of a target (i.e., the marshmallow) becomes relevant when the arm appears on the screen, because it is the association of the limb and the target to make the perceived action goal-directed. We thus divided each video into four Epochs: pre-hand (before the appearance of the limb, with target already present in the Grasping Hand condition), reaching (arm appears and complete the extension movement), withdrawing (arm has reached maximum extension and performs the bending movement) and post-hand (arm is no longer present).

Two repeated-measures ANOVA were conducted separately to analyze the time spent looking within a 3 visual degree radius window centered on the hand or target, with the factors of Condition (2 levels: Grasping Hand and Empty Hand) and Epoch (4 levels: pre-hand, reaching, withdrawing and post-hand). As for the time spent looking at the hand, it was observed that both main effects were significant (Condition: $F_{(1,7)} = 5.96$, $p = 0.045$, Epoch: $F_{(2.3, 16.1)} = 44.68$, $p < 0.001$), as well as their interaction (Condition*Epoch: $F_{(2.1, 14.7)} = 3.73$, $p = 0.047$ and $\eta^2_p = 0.347$). Post hoc comparisons with Bonferroni's correction revealed a significant difference between the Grasping (mean=0.179, $SE = 0.019$) and Empty Hand (mean=0.13, $SE = 0.006$, with $p = 0.02$) conditions once the hand has reached the target and retracts (withdraw epoch), with more time spent watching the hand in the case of goal-directed action. On the other hand, the

ANOVA conducted on the time spent observing the target revealed a significant effect of Epoch ($F_{(1.74, \ 15.7)} = 27.15$, $p < 0.001$) and of its interaction with Condition (Condition*Epoch: $F_{(1.4, \ 12.64)} = 4.4$ p = 0.046 and $\eta^2_p = 0.328$). Posthoc comparisons revealed a significant difference between the two conditions, this time inherent the reaching phase, with greater time spent looking at the target during the goal-directed action (mean = 0.135, SE = 0.025) than during the non-goal-directed condition (mean=0.074, SE = 0.011 with $p = 0.048$).

Altogether, these results (reported in Fig. 3) suggest basic saccade metrics are similar between the two experimental conditions, indicated by the absence of differences in the frequency and amplitude of the saccades. However, the gaze patterns do differ slightly. More specifically, the target and the limb attract gaze when on the screen simultaneously as part of a goal-directed action. This does not happen in the case of a non-goal-directed action.

## Discussion

Observing and understanding others' actions is crucial for social animals like primates. Numerous studies have shown that a fronto-temporo-parietal network mediates this function in Old World macaques and humans[1,5,7,14,15,47]. This so-called action observation network (AON) responds to goal-directed actions[6,7], and its main function seems to be to attribute a purpose to the actions of others in order to plan an effective response, providing the foundation for social interactions[29,48]. A strong conceptual overlapping exists between the definition of AON and that of the mirror neuron system, a brain network activated both during the observation and the execution of goal-directed actions[49–51]. While the AON is solely activated by action observation, the mirror neurons system is activated by both observation and execution of actions. Here, we investigated the existence of an AON in the common marmoset, a small New World primate that is becoming popular as powerful additional nonhuman primate model[52]. For this purpose, we presented awake marmosets with videos depicting a finalized action (grasping food) or a similar but non-goal-directed movement (grasping air) in an fMRI block-designed task, alternated by videos of their phase-scrambled versions.

The comparison between videos depicting an action (intact) and their phase-scrambled versions elicited strong activations in the occipital and temporal lobes, consistent with what we observed in previous studies[53,54] and overlapping with the location of body and face patches[43,44]. These activations, proceeding in a caudo-rostral direction, include portions of the visual areas V4/V4T, the fundus of the superior temporal sulcus (FST), the PGa-IPa and a wide part of the TE complex, in particular area TE3 and the most dorsal part of TEO. All these regions share similar cytoarchitectural features with the homologous macaque's areas[55]. Therefore, these patches seem to respond specifically to the presentation of a part of the body (the hand or the entire arm) in motion. A similar pattern of occipito-temporal activations has been reported during the presentation of videos depicting faces in marmosets[43,44,54], and low-level features of moving body-parts in macaques and human[11,22,45,56–62]. Within the AON, this node therefore seems to be responsible for the detection and processing of a biological movement, a fundamental step in the recognition of an external agent[45]. Only after this recognition is it possible to attribute an intention and a goal to the actions of the external agent. An intriguing aspect of our findings is the observed lateralization of activations, which displays a more extensive response network in the left hemisphere than the right. To our knowledge, there is no existing evidence for such lateralization in marmosets while observing moving body parts. One potential

explanation for this lateralization involves the composition of our sample, consisting of six right-handed marmosets and only one left-handed animal. Studies in humans have shown that right- and left-handers exhibit different interhemispheric activations in premotor and parietal regions during the observation of meaningless hand actions[63]. This disparity could emerge when comparing responses to moving body parts versus non-biological stimuli in motion, while the comparison between goal-directed and non-goal-directed actions might be less affected (thus accounting for the absence of this lateralization in our second comparison). However, the limited number of left-handed animals in our sample prevents us from verifying whether an effect of this nature exists in marmosets during action observation, leaving this hypothesis speculative. Moreover, the pattern of activations of our left-handed animal does not differ, from a qualitatively point of view, from the rest of the sample.

A second possible explanation relates to the signal quality acquired during our study. In a previous publication from our laboratory[64], we compared the temporal signal-to-noise ratio (tSNR) of the coils used in this study (chamber- and headpost-coil). While the newer headpost-coil exhibits improved tSNR compared to the previous chamber-coil, both display a stronger signal in the left hemisphere than in the right. Our functional imaging sequence, which uses a left-to-right acquisition direction, may have exacerbated this issue. However, this hypothesis does not account for the lack of such lateralization in the subsequent statistical comparison between goal-directed and non-goal-directed videos.

The main finding of this study comes from the comparison between goal-directed and non-goal-directed actions. As already observed by the first electrophysiological[49–51] and functional imaging studies in macaques[6,7,58,65] and humans[8,66,67], neurons in the AON show greater activity during the observation of goal- versus non-goal-directed movements. It has been hypothesized that this feature could provide the basis of the ability to recognize others' actions and understand their behavior and intentions[45,68].

While the existence of an AON has been established in both human and non-human Old World macaques[1,14,45,46,69], only a small number of neurons responding to goal-directed actions have been identified by a single neuron recording study in the ventrolateral prefrontal region of New World marmosets[39]. Thus, our study supports the existence of an extended AON in the common marmoset. This network shows strong correspondences with the AON in macaques and humans. The first (mirror) neurons in macaques responding to the observation of others' actions have been observed in areas F5 and 45[6,7,49–51], and regions with similar properties have also been found with fMRI in homologous brain areas in humans[5,17,30,66,67,70–74]. Our results show that large parts of the bilateral marmoset area 6 V (likely corresponding to macaque's area F5 and human area PMv) and of the ventrolateral prefrontal cortex (area 45 bilaterally, plus right areas 47 and 13, all corresponding to the homologous labeled areas in macaques) are part of this AON. The homologies between the AON of the three species continue with activations in the dorsal area 6 (6DR and 6DC, bilaterally), resembling what has been observed by electrophysiological[33,36,38] and functional imaging studies[30–32] in macaques and humans.

Outside the premotor and prefrontal cortex, one of the main AON regions is the STS in both humans[5,21,32,75] and macaques[7,19,60]. Here, we found a posterior region in the fundus of the temporal sulcus (FST) in the New World marmoset that was more active for the observation of goal-directed versus non-goal-directed movements, consistent with the findings of a macaque fMRI study that used a similar experimental paradigm[7]. In marmosets, this cluster extends also towards more lateral and inferior temporal regions, involving the more dorsal part of area

TE3 and the PGa-IPa area, confirming previous observations in macaques[58].

At the parietal level, the observation of goal-directed actions induces a bilateral cluster of activation including high-level visual areas (V3A, V6, V6A), the intraparietal areas MIP and LIP and the PG area. Although not part of the classic AON core, responses in these visual areas have previously been identified in macaques[24,65] and could be involved in the processing of visuospatial information required for the coding of reaching actions. Moreover, a circuit recruited in the planning and execution of arm reaching movements includes area V6A, area MIP and dorsal premotor area F2 (corresponding to area 6DC in marmosets)[76] in macaques. These regions are also activated bilaterally in our study and therefore could be a part of an extended AON. Further studies using different types of goal-directed actions and/or different effectors will be required to test this hypothesis.

Surprisingly, our study did not find activations at the level of the inferior parietal lobule and in area AIP, one of the hubs of the AON often reported in human[12,14,77–80] and macaque studies[9,14,25–27,81]. The rostral part of the IPL would indeed be the junction between the visual input coming from the STS and directed to the prefrontal and premotor cortices, structures not directly connected to each other in the macaque monkey[23,82–84]. Conversely, recent studies demonstrated a direct bidirectional connection between the ventrolateral prefrontal cortex, including area 6V, and area FST in common marmosets[39,85]. This could explain the reduced parietal activation for goal-directed actions in our study. However, the pattern of fronto-temporal anatomical connections of AIP seems to parallel the activations we observed for the action observation network, which is not the case for other parietal areas of the dorsal visual stream such as LIP, MIP or V6 (see Supplementary Note 2 and Supplementary Figure 3). This could position AIP as the communication hub between the dorsal visual stream and the action observation network. Nevertheless, the absence of activations during the presentation of goal-directed actions necessitates further investigation in this regard. A second possible explanation concern the content of the videos that we used: while humans and macaques are more accustomed to the use of the upper limbs for reaching, grasping and manipulating objects, marmosets often use their mouth to capture preys with limited avoidance behaviors, if left free to choose[86]. It is therefore possible that the use of videos depicting grasping-with-mouth actions may prove even more effective in mapping brain regions that respond to action observation in this primate species. Finally, another potential explanation may stem from the uncertain location of the anatomical boundary between the LIP and AIP regions in marmosets. The separation between the most ventral and dorsal parietal regions of the common marmoset is composed of a complex of areas likely homologous of the intraparietal areas of the macaques[55,87], but their functional distinction is still not well established. Evidence in support of this interpretation may be, for example, the fact that both AIP and LIP appear to be directly connected to MT (V5) area in marmosets[88], while in macaques MT projects directly to LIP, but not to AIP[89]. Both V5 and the portion of LIP at the border with AIP are part of the AON described by us in marmosets, while AIP, notoriously part of the AON of humans and macaques, is not part of it. Like the two previous hypotheses, this explanation also remains speculative, and further studies are desirable to explore this difference between the three species of primates.

In conclusion, our findings further support the existing literature on the human AON, specifically regarding subcortical activation at the thalamic level. Thalamic activation during the observation of goal-directed actions has been observed in both healthy humans[90,91] and patients with Parkinson's disease[92]. Errante and Fogassi, for instance, examined subcortical responses during the presentation and execution of meaningful actions and simple finger tapping actions[91]. The pulvinar activation they identified aligns with the subcortical activations reported in our study. Additionally, the marmoset AON encompasses a large bilateral portion of the anteroventral and anteromedial thalamus. To our knowledge, the involvement of this region has not been reported in the literature on the AON for either humans or macaques. Therefore, further studies using different types of actions or contexts are desirable to better understand this finding. Although the frontal and parietal regions reported in our study are homologous to those found in the AON in Old World macaques, some of them are also known to play a role in eye movements in marmosets[93–96]. However, the results of our eye-tracking study show that the frequency and amplitudes of saccades did not differ between the two main experimental conditions. Hence, it is highly unlikely that these activations are due to differences in eye movements between conditions. Moreover, a previous microstimulation study from our lab also showed facial and forelimb responses in this frontal region[96], and single neuron recordings have identified neurons involved in the generation and processing of vocalizations in these same frontal areas (45, 8Av, 6V, 6D) in marmosets[97]. Further electrophysiological recordings will be necessary to test whether separate neural populations perform these different functions or whether the same neurons have multiple response patterns (e.g. saccade, vocal-motor, or grasping responses). An interesting result comes from the epoch-by-epoch eye-tracking analysis, in which a significant difference is observed between the oculomotor patterns in the goal-directed and non-goal-directed conditions. In particular, our marmosets seem to focus more on the marshmallow in the Grasping Hand condition, compared to the target position of the Empty Hand one, although these two targets are largely spatially overlapped on the screen. This difference, however, only emerges during the reaching phase, when the arm appears and performs the extension movement. It is important to note that the target does not elicit any difference in the oculomotor behavior before that phase: when only a marshmallow (Grasping Hand, pre-hand phase) is represented on the screen or no target is presented (Empty Hand, pre-hand phase), the two experimental conditions do not differ. This result supports our fMRI findings, as it seems unlikely that the mere presence of a reward (i.e., the marshmallow), without inducing a specific oculomotor behavior, is capable of inducing the brain activations we reported. It is more likely that the attribution of a purpose to the action is due to the association of the limb and the target: while the time spent watching the hand during the reaching phase in the Grasping Hand and Empty Hand conditions does not differ, the greater time spent looking at the target in the Grasping Hand videos can explain the attribution of a purpose to such movement. This interpretation is in accordance with what previously observed in macaques[68]: the perception of the initial phase of a goal-directed movement is already able to activate the mirror neurons system, even when the final part of the action is obscured, but this happens only if, before the action, a target is presented. If, in the opposite case, the object is not shown (as in our Empty Hand condition), the mirror neuron system remains silent. Altogether, the results of the eye-tracking study therefore provide additional support to our fMRI results, confirming in the marmosets the presence of an action-observation network similar and comparable to that described in both humans and Old-World macaques.

In summary, our study provides the functional mapping of an extensive AON in the common marmoset. This network exhibits striking similarities with the well-characterized AON in humans and macaques, suggesting that the common ancestor of New and Old World primates possessed a system for understanding the actions of others. The identification of an AON in marmosets provides the foundation for targeted recording and stimulation studies that will advance our mechanistic understanding of primate social cognition.

## Methods

**Animal care and ethical approval.** All the experimental procedures here described were performed in accordance with the guidelines of the Canadian Council on Animal Care policy on the care and use of experimental animals and an animal use protocol #2021-111 approved by the Animal Care Committee of the University of Western Ontario.

**Subjects and experimental setup.** Seven common marmosets (*Callithrix jacchus*; three females, average age: 35.8 ± 8.5 months, ranging from 30 to 54 months, average weight: 398 ± 46.7 g, ranging from 328 to 462 g) took part in an awake fMRI study after the surgical implantation of an MRI-compatible head restraint chamber (*n* = 4, see ref. [98] for details) or of an MRI-compatible head post (*n* = 3). In order to reduce stress and anxiety due to scanning and to head-fixation, all marmosets underwent a 3-week training following the acclimatization procedure prior to the experiments (described in ref. [99]). All animals scored 1 or 2 on the behavioral assessment scale described by Silva and colleagues[100] after the training, reporting little or no signs of agitation and a calm behavior. All animals included in the sample had previous experience with the fMRI scanning setting, ranging from 4 to 30 fMRI sessions performed prior to this experiment.

Before MRI acquisition, the marmoset was placed in the sphinx position using the same MRI-compatible restraint system described by Schaeffer and collaborators[101]. After entering the animal holder, the monkey was firstly restrained using a neck and a tail plates. Taking advantage of either the MRI-compatible chamber or the head post, the head of the marmoset was then fixed to a five-channel (chamber) or eight-channel (head post) receive coil. A lubricating gel (MUKO SM1321N, Canadian Custom Packaging Company, Toronto, Ontario, Canada) was applied on the top of the chamber and on its edges in order to reduce magnetic susceptibility and artifacts. An MRI-compatible camera (model 12M-i, 60 Hz sampling rate, MRC Systems GmbH, Heidelberg, Germany) mounted on the front of the animal holder allowed the monitoring of the animal's condition and awake state by a veterinarian technician. Once in the scanner, marmosets faced a translucent plexiglass screen attached to the front of the scanner bore (~119 cm from animal's eyes) on which the visual stimuli projected by a SONY VPL-FE40 projector were reflected via a wall-mounted mirror. Maximum visual angle from the center to the side of the screen was 6.5°. Visual stimuli were presented via PowerPoint in synchronization with MRI TTL pulses triggered by a Raspberry Pi (model 3B + , Raspberry Pi Foundation, Cambridge, UK) running a custom python script.

**Task and stimuli.** Similarly to Nelissen and colleagues' studies[6,7], the visual stimuli used in this experiment consisted of movies showing reaching and grasping actions or movements of the upper limb. All videos were recorded inside the marmoset facility of the University of Western Ontario and included two different motor acts performed by the upper limb of a marmoset (none of the actors were included in the experimental sample nor were they housed in the same room as the fMRI experimental animals). Video editing was performed in Adobe Premiere Pro (Adobe, San Jose, California, USA). In the first experimental condition, hereafter called Grasping Hand (S1 and S2 movies), the movie represented the hand and forearm of a marmoset reaching towards and grasping a small piece of food (marshmallow, ~2 cm of diameter). In this condition, a complete goal-directed action is therefore presented in which the food is reached, grabbed and finally transported outside the scene. In the second experimental condition, hereafter called Empty Hand (S3 and S4 movies), the movement of the upper limb is very similar, but in this case the action is purposeless, as no object is present in the scene. To ensure the greatest similarity between the movements of the two conditions, Empty Hand videos were obtained using grasping movements towards a marshmallow, placed in the same position used in the Grasping Hand condition, removed just before the grasping phase. The marshmallow was then removed from the video in post-production. This approach enabled us to achieve a complete reaching-to-grasp movement that appeared goalless (as the target was not visible in the final video) while circumventing the challenge of training our marmosets to

perform an arm extension without a purpose. All videos lasted for 3 s; due to the speed of the recorded grasping movement, these clips were slowed down to fill this time interval (S1, S2, S3, and S4 movies are examples of slowed-down actions, slowing range 20–45%). Twelve different videos were originally recorded for the Grasping and the Empty Hand conditions. To check the effect of the direction of action, these 12 per-condition videos were mirrored, presenting rightward and leftward sequences intermingled within each run for each monkey (S1 and S3 show examples of leftward movements, S2 and S4 are examples of rightward movements). Nonetheless, this mirroring technique cannot be regarded as an efficient means to examine effects associated with the lateralization of the limb displayed in the video. The marmoset actors had the freedom to decide which arm to use for performing the movement in each video. Consequently, it is possible that two videos, both illustrating the same leftward movement, could feature a right arm and a left arm. In addition, to check possible effects due simply to the perception of moving stimuli, all videos were scrambled (see S5/S6 and S7/S8 movies for examples of scrambled grasping and empty hand conditions respectively) by random rotation of the phase information performed with a custom MATLAB script (MATLAB, The MathWorks, Natick, MA). The use of a constant seed made it possible to preserve the aspects of motion and luminance in scrambled videos, applying the same rotation matrix to each frame of each movie. To sum up, the visual stimuli used in our study included 24 videos per condition (Grasping Hand, Empty Hand, Scrambled Grasping Hand and Scrambled Empty Hand), 12 with a left-to-right reaching-to-grasp movement and 12 with the same movement performed in the opposite direction. Each run included 8 stimulation blocks (two blocks per condition), whose order was fully randomized across runs, sessions and animals. Each block was composed of a sequence of four 3-s videos belonging to the same experimental condition, for a duration of 12 s, and was interleaved 18 s of baseline, in order to allow the washing-out of the BOLD signal between experimental blocks (total duration of a run: 258 s). Sequences representing leftward and rightward movements were intermingled in the same run. The visual window occupied by the videos, centered on the screen, is a 25*14 cm (12°*6.7°) rectangle. The target (when present) fell always within the central visual field (min/max distance of the center of the marshmallow from the center of the screen: 0°/~3°). During the baseline, a black filled circular shape (1.5 cm of diameter, 0.72°) was presented at the center of the screen, but the animals were not required (nor trained) to fix: the only purpose of the black dot was to reduce the nystagmus sometimes present because of the ultra-high magnetic field, as previously observed in humans[102]. A diagram of the structure of a run can be seen in Fig. 4. The compliance of the animal during each run was checked and noted online by the investigator; runs in which the animal closed its eyes for two or more stimulation blocks (regardless of the experimental condition) were discarded from further analyses (N = 32). Animals did not receive any reward during the sessions.

**MRI acquisition.** All scanning sessions were carried out in a 9.4 T, 31 cm horizontal bore magnet (Varian/Agilent) and a Bruker BioSpec Avance III console with the software package Paravision-7 (Bruker BioSpin Corp), a custom-built high-performance 15-cm diameter gradient coil with 400-mT/m maximum gradient strength[103], and a five-[101] or eight-channel receive coil[64] at the Centre for Functional and Metabolic Mapping at the University of Western Ontario. An in-house built quadrature birdcage coil (12-cm diameter) functioned as transmit coil.

For functional imaging, gradient-echo-based, single-shot echo-planar images covering the whole brain were acquired over multiple daily sessions (TR = 1500 ms; TE = 15 ms; flip angle = 40°; FOV = 64 × 48 mm; matrix size = 96×128; voxel size = 0.5 mm isotropic; number of slices = 42 [axial]; bandwidth = 400 kHz; GRAPPA acceleration factor (left-right = 2)). To correct for spatial distortion, a second set of echo-planar images with the opposite phase-encoding direction (right-left) was collected. To perform anatomical registration, a T2-weighted structural image was acquired for each animal with the following parameters: TR = 7000 ms; TE = 52 ms; FOV = 51.20 × 51.20 mm; voxel size = 0.133 × 0.133 × 0.5 mm; number of slices =45 (axial); bandwidth=50 kHz, GRAPPA acceleration factor = 2.

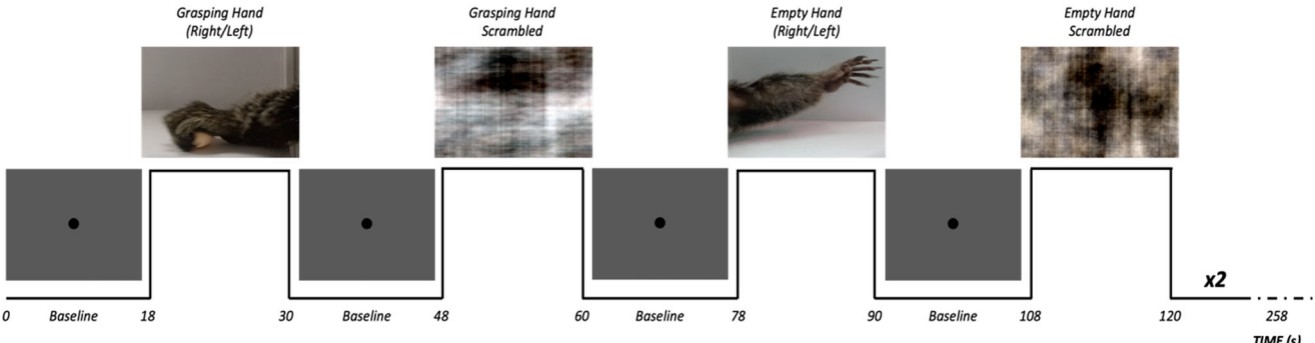

**Fig. 4 Experimental design for task-based fMRI.** In each run, 9 baseline blocks (18 s long) were alternated with experimental blocks (12 s long) of four different conditions: Grasping Hand, Empty Hand, Grasping Hand Scrambled, and Empty Hand Scrambled. Each run lasted 258 s globally.

Each session lasted approximately 50–60 min, including preparation of the animal and of the sequences. We acquired 10 functional runs that fulfilled our inclusion criteria for each marmoset, which required between 4 and 7 sessions, depending on monkey's compliance.

**Image preprocessing**. The data were preprocessed using a combination of AFNI[104] and FSL's functions[105]. The raw functional images were first converted to the NIfTI format using dcm2niix[106] and then reoriented (FSL's fslswapdim and fslorient) to correct the sphinx posture. The reoriented functional images were preprocessed through procedures for eliminating any outliers (detected via AFNI's 3dToutcount), despiking (AFNI's 3dDespike) and time shifting (AFNI's 3dTshift). The images thus obtained were registered to the base volume (extracted at the half of each run and therefore corresponding to the 86th volume) using AFNI's 3dvolreg function. Then, all volumes were smoothed (AFNI's 3dmerge, FWHM Gaussian kernel of 2 mm) and bandpass-filtered (AFNI's 3dBandpass, lowest frequency of 0.1 and highest frequency of 0.01). An average functional image was calculated for each run of each animal and then registered (FSL's FLIRT function) to the respective T2-weighted image. The transformation matrix thus obtained was subsequently used to carry out the 4D time series data. T2-weighted images were manually skull-stripped (removing the olfactory bulb) and the mask obtained was applied to the functional images.

Finally, anatomical and functional images were registered to the NIH marmoset brain atlas[107], using the Paxinos parcellation, via the nonlinear registration operated by ANTs' (Advanced Normalization Tools)[108] ApplyTransforms function. This atlas performed an initial manual delineation of 54 cortical and 16 subcortical areas taking advantage of multi-modal MRI contrasts, and refining it through the comparison with two histology-based atlases, the digital Paxinos atlas[109] from Marmoset Brain Architecture Project (http://marmoset.braincircuits.org/) and the digital Riken atlas[110] from BSINI Marmoset (http://brainatlas.brain.riken.jp/marmoset/).

**Eye tracking**. Marmosets have very large pupils which make video eye tracking challenging in the MR environment. Thus, the signal from the MRI-compatible camera used to assess the alert state of the animals in the scanner was not of sufficient quality to guarantee a reliable gaze analysis. To investigate potential differences in eye movements between our two conditions of interest (i.e., Grasping Hand and Empty Hand) we conducted an eye-tracking experiment outside of the scanner, in a sound attenuating chamber (Crist Instruments Co., Hagerstown, MD, USA) free of MRI-induced noise. 10 common marmosets (5 females, average age 35.5 ± 6.78 months, average weight 417.4 ± 57.7 g) took part in this experiment. Among them, 7 performed the eye-tracking task before ($n = 4$) or after ($n = 3$) the fMRI sessions (all the fMRI animals have thus been involved in the eye-tracking study). Depending on the type of implant (chamber or head-post), the animals were head restrained in the sphinx position in the MRI chair, following the same procedure described above, or upright, in a custom chair[98], inside the sound attenuating chamber. Grasping (24 videoclips for right- and leftward grasping) and Empty Hand videos (24 videoclips for rightward and leftward grasping) were presented with NIMH Monkeylogic[111] at 57 cm from the eye of the animal with the same size (in visual angle, thus reducing the dimensions of the window in which the videos were presented) described above (see "Task and stimuli"). At the beginning of each session, the eye position was calibrated by rewarding the monkey for 300 to 600 ms fixations on a 1-degree dot presented at the display center and at 6 degrees in each of the cardinal directions using NIMH Monkeylogic[111].

All stimuli were presented on a CRT monitor (ViewSonic Optiquest Q115, 76 Hz non-interlaced, 1600 × 1280 resolution). Eye position was digitally recorded at 1 kHz via video tracking of the left pupil (EyeLink 1000, SR Research, Ottawa, ON, Canada)[112].

**Statistics and reproducibility**. In the fMRI experiment, the scan timing was convolved to the BOLD response (AFNI's 3dDeconvolve) specifying the 'BLOCK' convolution and extracting a regressor for each condition (Grasping Hand, Empty Hand and the two Scrambled conditions) for each run to be used in the subsequent regression analysis. All the conditions were entered in the model, along with polynomial detrending regressors ($n = 5$). This regression generated four β-weight maps, corresponding to the four experimental conditions, per animal ($n = 7$) per run, that were registered to the NIH marmoset brain atlas[107]. These maps were thus compared through paired t tests (AFNI's 3dttest + +).

At the group level, the T-maps of each animal for each condition and for each run ($n = 10$) were converted into Z-maps. To investigate the neural responses induced by the perception of an intact biological movement versus a scrambled movement pattern, the Z score maps ($n = 140$) of the Grasping and Empty Hand conditions of each animal were compared via t test to those of the Scrambled Grasping Hand and Scrambled Empty Hand conditions. For protection against false positives, the results of this t-tests have been corrected with a minimum cluster-size resulting from 10000 Monte Carlo simulations (AFNI's 3dttest + + with Clustsim option, $p < 0.001$ and $\alpha = 0.05$, nearest-neighbor clustering method 2, two-sided).

Finally, the presence of areas selective for goal-directed actions was investigated by comparing the Z value maps ($n = 70$) of the Grasping Hand and Empty Hand

conditions. As described above, we compared these maps using a paired t test, which result was protected from false positives thanks to the same cluster-size correction (AFNI's 3dttest + + with Clustsim option, $p < 0.001$ and $\alpha = 0.05$, nearest-neighbor clustering method 2, two-sided).

In the eye-tracking experiment, eye movement analysis was conducted using an in-house written Python code. Eye velocity (visual degrees/second) was calculated through smoothing and numerical differentiation, and saccades were defined as eye movements with a radial eye velocity greater than 30 deg/s and amplitude greater than 0.35 visual degrees. To replicate the experimental conditions of the fMRI sessions, no reward was delivered during the eye-tracking. Overall differences in saccade frequency (number of saccades/second) and median amplitude were investigated through Wilcoxon signed-rank tests for two-paired samples. Furthermore, each video was divided into 4 epochs, based on the content: pre-hand (arm not yet present), reaching (arm appears and performs the reaching movement up to its maximum extension), withdrawing (arm has reached its maximum extension and performs the bending movement until it disappears from the screen) and post-hand (arm no longer present). In each of these phases, a window of 3 visual degrees was drawn around the hand, following its trajectory (in the pre-hand and post-hand phases the window includes the portion to the far right or left of the screen, depending on the type of video—leftward or rightward reaching). A second window of 3 visual degrees was also drawn around the target: the marshmallow in the Grasping Hand condition and the position reached by the hand following the maximum extension of the arm in the Empty Hand condition. This last window, like the previous one, followed the position of the target during the whole video (in the Empty Hand condition, the target window coincided with the hand window starting from the withdrawing phase). In this way, it was possible to compute the proportion of time spent watching the hand or target during the four epochs. The differences between the different epochs and/or the two experimental conditions were then analyzed through two distinct repeated-measures ANOVA (one for the visual window centered on the hand, one for the centered on the target) with the factors: Condition (Grasping Hand vs Empty Hand) and Epoch (pre-hand, reaching, withdrawing, post-hand). Where necessary, violations of sphericity were corrected by the Greenhouse–Geisser correction. Any significant effects were further investigated through post-hoc comparisons with Bonferroni correction.

**Reporting summary**. Further information on research design is available in the Nature Portfolio Reporting Summary linked to this article.

## Data availability
Data and videos supporting this study are available on OSF at https://osf.io/hvbmy/?view_only=6ea57106e8ce464fb0574a1acbc2f89d.

## Code availability
All the codes supporting this study are available on OSF at https://osf.io/hvbmy/?view_only=6ea57106e8ce464fb0574a1acbc2f89d and on Zenodo (https://doi.org/10.5281/zenodo.7877614).

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

## Acknowledgements

We wish to thank Cheryl Vander Tuin, Whitney Froese, Hannah Pettypiece, and Miranda Bellyou for animal preparation and care; Dr. Alex Li for scanning assistance; Dr. Kyle Gilbert and Peter Zeman for coil designs; and Dr. Kevin Johnston for video footage assistance and his careful reading of this manuscript.

## Author contributions

A.Z.: conceptualization, methodology, formal analysis, writing—original draft, writing—review and editing. A.D.: methodology, formal analysis, writing—review and editing. J.S.: formal analysis, writing—review and editing. S.E.: conceptualization, methodology, funding acquisition, supervision, writing—review and editing.

## Competing interests

The authors declare no competing interests.
