## [Peer Review File · Communications Biology]

Reviewers' comments:

Reviewer #1 (Remarks to the Author):

Review of Zanini et al. Comms Biol

This paper fills an important gap in the literature about cortical organization in new world monkeys. Given the increasing use of marmosets in neuroscience research, it is important to understand what differences the smaller brain size imposes relative to macaques and humans, in terms of cortical physiology. The paper reports on the existence of an action observation network (AON) in marmosets, and concludes that this network includes areas in similar locations as in previously studied primates. This result opens the way for the use of marmosets for studying some of the most important characteristics of primate brains.

The paper is clearly written and has no major technical or conceptual flaws. The conclusions are original, and the (limited) previous literature on the topic is appropriately cited. My comments are relatively minor.

1. One aspect of the data that deserve additional discussion is the laterality of the responses, particularly in Figure 1. I did not expect that much interhemispheric difference. Maybe the authors could discuss if there is any previous evidence of highly lateralised responses in marmosets.
2. Page 13 indicates that mirror videos were used, but it would be important to clearly state if these would represent both right and left hands, as seen by the observer. I would go one step further and analyse the data separately for videos showing left hand versus right hand movements, to try to get further insight on the possibility that laterality effects can be at least explained according to which hand is being seen.
3. Another aspect that requires documentation and discussion is the amount of variability in results. Experiments were done in 7 marmosets, but it is unclear if there was variation. For example, were the laterality effects consistently observable, on an animal by animal basis? Were the same areas consistently activated across cases, and if not, which areas were?
4. There is a substantial literature about hand preferences in marmosets, mostly by the Lesley Rogers group (e.g. Prog Brain Res. 2018;238:91-113), which could be discussed. If the hand preference of each animal was known, then including analyses of left-handed, right-handed and bilateral animals would be even more relevant. If this was not the case, then this could be mentioned as a limitation and suggestion for future work.
5. The lack of activation in AIP is interesting, and the two interpretations provided appear viable. However a third possibility is that the border between LIP and AIP in marmosets has not been completely well established. Have there been any functional studies that confirm the location of this border? If not, maybe this could be flagged as something for future work.
6. As a suggestion of additional analysis related to 4, above: given the distribution of activation linked to the AON established by the present study, how does this relate to the anatomical connections of the above parietal areas (where known)? Is it the case that the distribution of haemodynamic signals parallels the anatomical and/ or functional connectivity of any of the putative marmoset dorsal stream areas such as V6, MIP, LIP and V6a?
7. More detail about the registration process would be desirable. Which parcellation of cortex was used, and to what extent are these areas comparable to those in the macaque monkey? Reference 94 is to an MRI atlas but there is no information about how these areas were established (based on comparative architectonics?). Particularly for the areas considered to be key in the AON, additional

information regarding what is known about them needs to be provided, perhaps in the Discussion.

8. The existence of activation foci in subcortical structures is mentioned in the Results, but not discussed. Have corresponding AON regions been also described in macaques and humans? What is known about the possible function of the anteromedial thalamus in the context of action observation?

Reviewer #2 (Remarks to the Author):

This is a first study of brain activations in the marmoset in response to observation of goal-oriented reach behavior, as previously studied in humans and macaque monkeys. Marmosets are provided videos to observe in a 9.4T MRI. Videos consist of 'goal-directed (Grasping hand)' and 'non goal-directed (Empty hand)' grasp. Comparison of the fMRI images of goal-directed vs non goal-directed revealed primary activations in the prefrontal (areas 6, 8, 47, 11, 13), dorsomedial prehension circuits (areas MIP, LIP, PG, V3A, V6), some regions in V1, V2, V4, and tempo-occipital (areas V4, V4T, V5, TEO, FST, PGa-IPa, TE3), retrosplenial (30), and thalamic (pulvinar, LD, anteroventral) areas. These areas are largely similar to previously described action observation networks in macaques and humans. This study shows that it is possible to study behaviorally-related brain activations in marmosets in ultrahigh field MRI, further helping to establish marmosets as a model for brain studies. Some concerns are listed below.

Major comments:

1. The interpretation that the Empty hand is non goal-oriented needs some more support. Was there a different target or an internally generated goal? Was there a goal but the target was missing? When does an animal extend the arm/hand without purpose?

2. The conclusion that there is no difference between eye movements during the goal-directed vs non goal-directed videos needs additional analysis. Eye movement traces need to be further quantified as current analyses are a bit simplistic. Saccade frequency and amplitude are important aspects of eye movements but do not characterize the locations or distributions of eye movements, something which is relevant to the importance of the goal in each condition. Neither does it monitor internal tracking of a goal prior to or expected behaviors after acquisition of the goal. For example, it might be expected that during goal oriented behavior, more eye movements towards the goal or prior to achieving the goal would occur, or there might be covert attention towards the location of the goal.

3. Are there any goal-directed behaviors that are unique to marmosets? This might add to novelty of the study.

4. Example of one set of 8 videos should be provided

Other comments

1. Some basic things are missing. I would like to see some BOLD signal timecourses, single condition activation maps, and eye movement traces.

2. Please clarify: the 8 videos are Grasping Hand, Empty hand, Scrambled Grasping Hand, Scrambled Empty Hand, and the mirror reversals of these videos?

3. Are there differences in pupillary dilation in goal-oriented vs non goal-oriented tasks?

4. Where is the target in visual field?

5. UHF causes nystagmus? Does this cause discomfort to the animal?

6. Please provide a reference to the behavior 'grasp with mouth'

Reviewer #3 (Remarks to the Author):

In this study, 7 awake common marmosets (*Callithrix jacchus*) were presented with videos showing the arm and hand of another marmoset making movements to either a visible goal (grasping a piece of marshmallow) or an invisible goal (same videos with the marshmallow deleted making the hand look as if it was grasping air). Brain responses to these experimental conditions were examined at a magnetic field strength of 9.4 T using an fMRI block-design alternating the actual videos and their scrambled versions for ten runs per animal, that is, a total of 70 runs. This was completed by an out-of-the-scanner eye-tracking experiment in 10 monkeys confirming that Reaching & Grasping Food and Reaching & Grasping Air videos elicited similar oculomotor patterns.

New World marmosets have long attracted neuroscientists' interest. Prosocial, they might more closely approximate human cooperative social behavior than the competitive Old World macaques having heretofore dominated nonhuman primate neuroscience. This alternative primate model will likely gain even further importance in the future given the dire shortage of lab macaques that plagues worldwide research in the wake of Covid19 pandemic.

High-field fMRI is a state-of-the-art technique mastered by few neuroscientists. The methods used to collect and analyze imaging data should thus be evaluated by an expert. I am not competent to critically evaluate the authors' choices. Otherwise the paper is well-written easy to read, the behavioral methods are sound and the imaging results are clearly described and illustrated. I have but only minor comments and questions.

1. Could the authors provide information about the monkeys' prior experience? Were they naïve or on the contrary highly familiar with fMRI scanning?
2. Monkeys received no reward to look at the videos during imaging and eye tracking. Does it mean that, unlike macaques, marmosets do not need any water or food control to comply with testing? Did they steadily look at stimuli while in the scanner? How many runs had to be discarded out the 70 acquired runs?
3. Rather than "Old World primates" I would use "Old World macaques" to be more accurate.
4. I may have missed it but I could not find a link to samples of the videos.

Point-by-point response to referees' comments

Reviewer #1 (Remarks to the Author):

Review of Zanini et al. Comms Biol

This paper fills an important gap in the literature about cortical organization in new world monkeys. Given the increasing use of marmosets in neuroscience research, it is important to understand what differences the smaller brain size imposes relative to macaques and humans, in terms of cortical physiology. The paper reports on the existence of an action observation network (AON) in marmosets, and concludes that this network includes areas in similar locations as in previously studied primates. This result opens the way for the use of marmosets for studying some of the most important characteristics of primate brains.

The paper is clearly written and has no major technical or conceptual flaws. The conclusions are original, and the (limited) previous literature on the topic is appropriately cited. My comments are relatively minor.

We thank the reviewer for taking the time to carefully analyze our manuscript and study. The points raised regarding our results and their interpretation are insightful, and we have expanded and modified the manuscript to address these comments. Please find below, point-by-point responses to the reviewer's comments.

1. One aspect of the data that deserve additional discussion is the laterality of the responses, particularly in Figure 1. I did not expect that much interhemispheric difference. Maybe the authors could discuss if there is any previous evidence of highly lateralised responses in marmosets.

We concur with the reviewer's astonishment about the unanticipated lateralization, made even more remarkable as it is absent in the comparison between the primary experimental conditions (Figure 2). This imbalance appears to be specifically linked to the comparison between videos portraying "intact" movements and phase-scrambled videos. To our knowledge, there is no prior evidence of such pronounced lateralized responses in marmosets for this type of analysis.

One hypothesis we propose relates to our sample composition: among the seven animals in our sample, six are right-handed, and only one is left-handed. This manual preference may underlie the lateralization in processing observed body parts when compared to videos showing non-body stimuli in motion. However, we have not found any supporting evidence in marmosets for this explanation, and the limited number of left-handed marmosets in our sample hinders further exploration of this idea. Willems and Hagoort (2009, Brain Research) observed that right- and left-handed healthy humans exhibit distinct interhemispheric activations when viewing meaningless hand actions. However, their research primarily focuses on premotor and parietal regions, leaving us unable to confirm if a similar phenomenon occurs in the marmoset's occipito-temporal area.

Another hypothesis pertains to the signal quality obtained during the fMRI sessions. In a prior publication from our laboratory (Gilbert et al., 2023), we compared the temporal signal-to-noise ratio (tSNR) of the coils used in this study (chamber-coil and headpost-coil). Although the newer headpost-coil exhibits superior tSNR compared to the chamber-coil, both display a stronger and less disrupted signal in the left hemisphere than in the right hemisphere (refer to the attached figure). Our functional imaging sequence employs a left-to-right acquisition direction, which may have exacerbated this issue.

From: Gilbert, K. M., Dureux, A., Jafari, A., Zanini, A., Zeman, P., Menon, R. S., & Everling, S. (2023). A radiofrequency coil to facilitate task-based fMRI of awake marmosets. *Journal of Neuroscience Methods*, 383, 109737.

We thus modified the main text to discuss better this aspect. Now, from page 11 (line 213) to 12 (line 233) is possible to read:

An intriguing aspect of our findings is the observed lateralization of activations, which displays a more extensive response network in the left hemisphere than the right. To our knowledge, there is no existing evidence for such lateralization in marmosets while observing moving body parts. One potential explanation for this lateralization involves the composition of our sample, consisting of six right-handed marmosets and only one left-handed animal. Studies in humans have shown that right- and left-handers exhibit different interhemispheric activations in premotor and parietal regions during the observation of meaningless hand actions. This disparity could emerge when comparing responses to moving body parts versus non-biological stimuli in motion, while the comparison between goal-directed and non-goal-directed actions might be less affected (thus accounting for the absence of this lateralization

in our second comparison). However, the limited number of left-handed animals in our sample prevents us from verifying whether an effect of this nature exists in marmosets during action observation, leaving this hypothesis speculative.

A second possible explanation relates to the signal quality acquired during our study. In a previous publication from our laboratory⁶¹, we compared the temporal signal-to-noise ratio (tSNR) of the coils used in this study (chamber- and headpost-coil). While the newer headpost-coil exhibits improved tSNR compared to the previous chamber-coil, both display a stronger signal in the left hemisphere than in the right. Our functional imaging sequence, which uses a left-to-right acquisition direction, may have exacerbated this issue. However, this hypothesis does not account for the lack of such lateralization in the subsequent statistical comparison between goal-directed and non-goal-directed videos.

2. Page 13 indicates that mirror videos were used, but it would be important to clearly state if these would represent both right and left hands, as seen by the observer. I would go one step further and analyse the data separately for videos showing left hand versus right hand movements, to try to get further insight on the possibility that laterality effects can be at least explained according to which hand is being seen.

We thank the reviewer for the interesting suggestion. As he pointed out, all the videos were mirrored, thus producing video clips representing both right and left arms/hands performing a reaching movement. However, in the videos we recorded the animal "actors" were free to use their right or left arm to reach for the marshmallow, according to their preference. It is therefore possible that two videos, both representing a reaching movement from right to left, depict two opposite hands. Furthermore, the sequences we use are made up of 4 of these video clips, not taking into account which arm is depicted in the video. Because of our experimental design, it is thus difficult to establish the lateralization effects depending on which limb is observed. We mirrored all the videos to try to balance the possible differences related to these aspects of lateralization (both lateralization of the movement and of the limb shown). However, we fully agree with the reviewer regarding this aspect, which we believe is an interesting starting point for future experiments, possibly testing a sample including a similar number of right-handed and left-handed animals, in order to investigate the lateralization of the movement perceived in relation to the hand preference.

We clarified the issue of video lateralization more in our experiment; the main text, on page 18 (starting from line 414), now reports:

"Nonetheless, this mirroring technique cannot be regarded as an efficient means to examine effects associated with the lateralization of the limb displayed in the video. The marmoset "actors" had the freedom to decide which arm to use for performing the movement in each video. Consequently, it is possible that two videos, both illustrating the same leftward movement, could feature a right arm and a left arm.."

3. Another aspect that requires documentation and discussion is the amount of variability in results. Experiments were done in 7 marmosets, but it is unclear if there was variation. For example, were the laterality effects consistently observable, on an animal by animal basis? Were the same areas consistently activated across cases, and if not, which areas were?

We concur with the reviewer that the main text does not provide an explanation for this aspect. The limited statistical power of the individual-level analyses precludes us from deriving reliable quantitative interpretations. However, it offers valuable information that supports the results presented in the main text. Consequently, we have added a Supplementary Information section to our manuscript, detailing individual activations for the two primary contrasts (Intact vs Scrambled movement and Grasping Hand vs Empty Hand). Additionally, we generated overlap maps for both contrasts based on these individual results. After conducting t-tests at the individual level, we created a mask for each marmoset, assigning the value 1 to voxels with a z-score greater than 1.96 and the value 0 to all other voxels. By summing the seven masks, we generated a map displaying the "frequency" of activation for each voxel. Voxels activated by the same contrast in all marmosets within the sample will have a value of 7, while voxels never activated will have a value of 0. These results allow us to 1) assess the variability of individual outcomes and 2) identify the most frequently activated brain areas for each of our contrasts.

We have further elaborated on this point in the Supplementary Information, dedicating a specific section and introducing two new figures to address the reviewer's comments.

Upon examining these figures, we cannot attribute a "pervasive" lateralization of activations for both contrasts. Although some animals display primarily left hemisphere activation patterns (e.g., M5 in Intact vs Scrambled Movement, Figure S1-C), the pattern appears bilateral for most of our sample. However, panel C of Figure S1 reveals more extensive overlapping zones in the left hemisphere than the right, despite the latter containing an overlap "peak" located at the border between the FST, PGa-IPa, and T3 areas. This observation could potentially explain the lateralization noted in the main text: individual activations in the left hemisphere overlap to a greater extent, but fewer marmosets share the same activations. Conversely, the right hemisphere has more restricted overlap areas, but these areas concentrate more individual activations.

Regarding consistently activated areas across cases, overlap peaks of 4-5 marmosets are found for both contrasts. In the comparison between Intact and Scrambled movement, the most frequently activated regions encompass a bilateral occipito-temporal cluster (FST, PGa-IPa, and T3) and a prefrontal/premotor cluster primarily in the left hemisphere (areas 45, 6Va). For the contrast between Grasping and Empty Hand, the most overlapping occipito-temporal cluster resembles that of the previous contrast but extends further into the ventral part of the V4T area. At the prefrontal level, a more extensive bilateral overlap is observed, including the dorsal and ventral premotor areas (6DC, 6DR, and 6Va) and the prefrontal areas 45, 47, 8c, and 8Av.

We thus modified the manuscript accordingly, including a new Supplementary Information section and adding references in the main text.

Page 4, line 93: See Supplementary Information for individual maps.

Page 7, line 113: See Supplementary Information for individual maps.

From Page 37: **Supplementary Information**

Individual results and overlapping across animals

To more thoroughly represent the variability of activations on an individual basis, Figures S1-C and S2-C display the individual maps of the seven marmosets in our sample for the two main experimental contrasts of our study (Intact vs Scrambled Movement and Grasping vs Empty Hand, respectively). By creating an individual mask that includes only voxels with a z-score > 1.96 ($p < 0.05$ uncorrected) and summing the seven masks obtained using AFNI's 3dcalc function, we can observe the extent of overlap among individual results. This overlap is represented in panels A (left hemisphere) and B (right hemisphere) of Figures S1 (Intact vs Scrambled Movement) and S2 (Grasping vs Empty Hand). In this representation, a voxel active for a specific contrast in all marmosets will have a value equal to 7, while a voxel never active will have a value of 0. Although the statistical power at the individual level is too low for quantitative analysis, this representation provides information that supports and extends previous observations.

For instance, in the contrast between Intact and Scrambled movement, Figure S1 demonstrates that the left hemisphere activation lateralization is not consistent across all tested animals. While this pattern is reproduced in some animals (particularly M4 and M5), the extension and amplitude of activations in most marmosets appear comparable between the two hemispheres. Additionally, a bilateral prefrontal activation cluster similar in localization to that found in the comparison between goal-directed and non-goal-directed actions can be observed in M2, M4, M6, and M7. However, this cluster does not withstand the stricter statistical correction performed on the entire sample. Panels A and B of Figure S1 confirm the presence of this prefrontal cluster, supporting the main text and Figure 1 observations: the greater overlap of individual activations is found at the prefrontal (particularly in areas 45 and 6Va) and occipito-temporal levels (with overlapping peaks in areas FST, PGa-IPa, and TE3), bilaterally.

Similarly, the individual maps of the goal-directed vs non-goal-directed actions comparison provide supporting information for previous descriptions. Figure S2-C reveals that, in this case, the prefrontal cluster is more extensive and consistent across monkeys than in the previous contrast. This greater stability bolsters the involvement of this region in the marmosets' AON. In panels A and B, the overlapping regions of individual results are more extensive, indicating greater consistency among these maps. Overlap peaks in the prefrontal region include the dorsal (6DR and 6DC) and ventral premotor areas (6Va), as well as prefrontal areas 45, 47, 8C, and 8Av. Posteriorly, activations of 5 out of 7 animals overlap in the left FST's most caudal part, but bilaterally, overlap of 3 or 4 marmosets can also be observed in the PGa-IPa, TE3, and V4T areas.

Intact vs Scrambled Movement

Individual overlapping

Figure S1. Individual maps and their overlapping for the Intact vs Scrambled Movement comparison. In panel C, the results of the individual t-tests comparing Grasping Hand + Empty Hand videos versus Scrambled Grasping Hand + Scrambled Empty Hand ones. Results are reported at $p < 0.05$ uncorrected. Panels A (left hemisphere) and B (right hemisphere) show the overlap between the 7 individual maps: a value of 1 means that the voxel is activated only for one monkey, whereas a voxel activated by the same contrast in all monkeys will report a value of 7. White lines delineate the cerebral areas included in the Paxinos parcellation of the NIH marmoset brain atlas (Liu et al., 2018).

Figure S2. Individual maps and their overlapping for the Grasping Hand vs Empty Hand comparison. In panel C, the results of the individual t-tests comparing Grasping Hand versus Empty Hand videos. Results are reported at $p < 0.05$ uncorrected. Panels A (left hemisphere) and B (right hemisphere) show the overlap between the 7 individual maps: a value of 1 means that the voxel is activated only for one monkey, whereas a voxel activated by the same contrast in all monkeys will report a value of 7. White lines delineate the cerebral areas included in the Paxinos parcellation of the NIH marmoset brain atlas (Liu et al., 2018).

4. There is a substantial literature about hand preferences in marmosets, mostly by the Lesley Rogers group (e.g. Prog Brain Res. 2018;238:91-113), which could be discussed. If the hand preference of each animal was known, then including analyses of left-handed, right-handed and bilateral animals would be even more relevant. If this was not the case, then this could be mentioned as a limitation and suggestion for future work.

We thank the reviewer for drawing our attention to this important aspect. The representation of the results at the individual level used to respond to the previous comment also allows us to investigate the possible differences between right- and left-handed marmosets. Unfortunately, only one of our animals showed a preference for using the left hand, while the remaining six were right-handed. The analysis of differences, given this imbalance, can only be made on a qualitative level, but we agree with the reviewer that further studies in this regard are desirable.

The pattern of activations of M3 (the only left-handed animal) for both contrasts performed does not seem to show obvious differences compared to what can be observed for right-handed animals. In the contrast between Intact and Scrambled movement, for example, the extension of the activations at the temporal level seems to be equivalent between the two hemispheres, and comparable to that shown by M4, a right-handed animal. A possible difference could be the lateralization of the activations at the orbitofrontal level: while most animals show consistent bilateral activations between the two hemispheres, M3 shows a lateralized activation on the left. However, it is impossible to say if this is due to manual preference, and M5, right-handed, shows a small orbitofrontal activation cluster only in the right hemisphere. Even in the Grasping vs Empty Hand comparison, M3 shows no significant differences from the rest of the sample: the localization and extension of the activations at the occipito-temporal level is consistent with what reported by right-handed animals, and the reduced prefrontal activation is comparable to that of some other marmosets (M4, M5 left hemisphere or M6 right hemisphere).

We have therefore modified the main text to deepen this aspect and have devoted a specific discussion in the Supplementary Information, after the description of the individual maps.

Page 11, lines 226-227: Individual results for right- and left-handed animals and a qualitative discussion are reported in Supplementary Information.

Page 38, from line 877:

An interesting aspect, but that we cannot investigate in depth because of the composition of our sample, concerns the manual preference shown by our marmosets. As already amply demonstrated in literature¹⁰⁹⁻¹¹¹, marmosets often show a strong manual preference in tasks of reaching for food, and such preference may reflect hemispheric dominance of other cognitive domains¹¹². Unfortunately, only one (M3) of the 7 animals we tested shows a preference for the use of the left hand, and we are therefore unable to investigate statistically differences in the AON related to this preference. In addition, the lack of separation of videos representing actions carried out with the left hand and with the right hand in our experimental design adds a degree of uncertainty to any possible interpretation. On a purely qualitative level, the Figures S1-C and S2-C do not seem to report a different pattern of activations for M3 compared to the 6 right-handed animals.

The impossibility to test the relationship between the manual preference of marmosets and the extension/lateralization of their action observation network is therefore a limit of this study and a possible starting point for future work.

5. The lack of activation in AIP is interesting, and the two interpretations provided appear viable. However a third possibility is that the border between LIP and AIP in marmosets has not been completely well established. Have there been any functional studies that confirm the location of this border? If not, maybe this could be flagged as something for future work.

We appreciate the reviewer's suggestion on this matter. To date, we are not aware of any additional studies that have investigated and better delineated the functional separation between the intraparietal areas of common marmosets. In our study, we utilized the parcellation by Paxinos and colleagues (Paxinos et al., 2012), in which the separation between different intraparietal areas is described as "a complex of areas likely homologous" to their counterparts in macaques. Even in Rosa et al., 2009, a more specific and detailed separation between AIP and LIP in marmosets is not identifiable. We recognize that the AIP region exhibits differences between marmosets and macaques, such as a direct connection between AIP and V5 in marmosets (Abe et al., 2018) that is absent in macaques (Blatt et al., 1990). The hypothesis suggested by the reviewer is thus highly plausible, and we have updated the main text to include it among our possible explanations.

Page 14, line 291-301:

Finally, another potential explanation may stem from the uncertain location of the anatomical boundary between the LIP and AIP regions in marmosets. The separation between the most ventral and dorsal parietal regions of the common marmoset is composed of a complex of areas likely homologous of the intraparietal areas of the macaques^{52,84}, but their functional distinction is still not well established. Evidence in support of this interpretation may be, for example, the fact that both AIP and LIP appear to be directly connected to the MT (V5) area in marmosets⁸⁵, while in macaques the MT area projects directly to LIP, but not to AIP⁸⁶. Both V5 and the portion of LIP at the border with AIP are part of the AON described by us in marmosets, while AIP, notoriously part of the AON of humans and macaques, is not part of it. Like the two previous hypotheses, this explanation also remains speculative, and further studies are desirable to explore this difference between the three species of primates.

6. As a suggestion of additional analysis related to 4. above: given the distribution of activation linked to the AON established by the present study, how does this relate to the anatomical connections of the above parietal areas (where known)? Is it the case that the distribution of haemodynamic signals parallels the anatomical and/or functional connectivity of any of the putative marmoset dorsal stream areas such as V6, MIP, LIP and V6a?

We thank the reviewer for their suggestion. Following their advice, we have added a paragraph in the supplementary materials discussing the comparison between our Grasping Hand vs Empty Hand activations and the anatomical and functional connectivity of the dorsal visual stream areas, specifically V6, V6A, LIP, MIP, and AIP extension. This analysis reveals

potential evidence supporting the inclusion of AIP in the action observation network of common marmosets. By examining Figure S3 (panel B), AIP is the only area in the dorsal visual stream that shows a direct anatomical connection with the frontal areas we included in the AON, particularly area 45 and the premotor region. Although all the areas analyzed in this study display functional connectivity that may parallel the fronto-temporo-parietal activations of the common marmoset AON, only injections into AIP revealed anatomical connections with the most frontal part of this network. This parallelism, along with the absence of a clear activation of AIP in our Grasping Hand vs Empty Hand contrast, raises an important question regarding the inclusion of this area within the AON. Further studies are needed to clarify this point, investigating whether AIP may serve as the parietal node connecting the dorsal visual stream to the action observation network.

The main text has been modified accordingly and a supplementary section has been added:

Page 14, lines 280-285:

However, the pattern of fronto-temporal anatomical connections of AIP seems to parallel the activations we observed for the action observation network, which is not the case for other parietal areas of the dorsal visual stream such as LIP, MIP or V6 (see Supplementary Information). This could position AIP as the communication hub between the dorsal visual stream and the action observation network. Nevertheless, the absence of activations during the presentation of goal-directed actions necessitates further investigation in this regard.

Page 41:

Intrigued by the lack of activation in AIP within the action observation network of common marmosets, we conducted a qualitative comparison between the anatomical-functional connections of the dorsal visual stream areas (i.e., V6, V6A, LIP, MIP, and AIP) and the network of activations reported through the contrast between Grasping Hand and Empty Hand. Figure S3 displays A) the action observation network of the common marmoset, B) the anatomical connectivity of the dorsal visual stream areas (Marmoset Brain Connectivity Atlas^{106,113}) and C) the functional connectivity of these areas (Marmoset Connectome¹¹⁴). What emerges from this analysis may represent further evidence in favor of the inclusion of AIP in the action observation network of the common marmosets. Observing figure S3 (panel B), in fact, AIP is the only area of the dorsal visual stream to report a direct anatomical connection with the frontal areas included by us in the AON, and in particular with area 45 and the premotor region. While all the areas we analyzed exhibit functional connectivity that may parallel the fronto-temporo-parietal activations of the common marmoset AON, only injections into AIP revealed anatomical connections with the most frontal part of this network. This parallelism, combined with the absence of clear activation of AIP in our Grasping Hand vs Empty Hand contrast, raises important questions about the inclusion of this area within the AON. Further research is necessary to determine whether AIP could serve as the parietal node connecting the dorsal visual stream to the action observation network.

Figure S3. Anatomical and functional connectivity of the areas of the dorsal visual stream in the common marmoset. In panel A, the action observation network, as obtained contrasting our Grasping Hand vs Empty Hand conditions. Results (z-score) are reported at $p < 0.001$ and cluster-size corrected with a Monte-Carlo method. In panel B, the anatomical connections of the areas included in marmoset's dorsal visual stream, obtained through injections of anterograde tracers and available online (Marmoset Brain Connectivity Atlas^{108, 109}). In panel C, the functional connectivity of these regions, obtained from fully awake marmosets and available online (Marmoset Connectome¹¹⁰).

7. More detail about the registration process would be desirable. Which parcellation of cortex was used, and to what extent are these areas comparable to those in the macaque monkey? Reference 94 is to an MRI atlas but there is no information about how these areas were established (based on comparative architectonics?). Particularly for the areas considered to be key in the AON, additional information regarding what is known about them needs to be provided, perhaps in the Discussion.

We agree with the reviewer regarding the need for some further information concerning the MRI atlas that we used for recording and for representing/discussing our results. The atlas we used (described in Liu et al., 2018, NeuroImage) was built starting from multi-modal MRI contrasts, which include T2 weighted images, magnetization transfer ratio (or MTR) and diffusion MRI images. These contrasts reflect different aspects of the brain tissue (i.e., myelination, macromolecules, orientation of diffusion), which allow this atlas to achieve a high level of consistency with histological atlases (such as Paxinos or Riken atlas). The reported parcellation is therefore multidimensional, but the labeling of the different brain areas was carried out trying to report the maximum consistency with the Paxinos atlas (Paxinos, 2012). In this last, regions of the marmoset's brain that share the cytoarchitectural characteristics of a specific area of the macaque's brain were given the same label. It is therefore possible to state that, although the MRI atlas of Liu and colleagues is multidimensional and not only based on the cytoarchitectural structure of the brain tissues, the same labels used in the parcellation of the macaque's brain have been applied to the marmoset's brain, wherever possible.

We modified the main text accordingly:

Page 11, lines 206-207:

All these regions share similar cytoarchitectural features with the homologous macaque's areas⁵².

Page 13, lines 249-252:

Our results show that large parts of the bilateral marmoset area 6V (likely corresponding to macaque's area F5 and human area PMv) and of the ventrolateral prefrontal cortex (area 45 bilaterally, plus right areas 47 and 13, all corresponding to the homologous labeled areas in macaques) are part of this AON.

Page 20, lines 479-485:

Finally, anatomical and functional images were registered to the NIH marmoset brain atlas¹⁰¹, using the Paxinos parcellation, via the nonlinear registration operated by ANTs' (Advanced Normalization Tools)¹⁰⁵ ApplyTransforms function. This atlas performed an initial manual delineation of 54 cortical and 16 subcortical areas taking advantage of multi-modal MRI contrasts, and refining it through the comparison with two histology-based atlases, the digital Paxinos atlas¹⁰⁶ from Marmoset Brain Architecture Project (<http://marmoset.braincircuits.org/>) and the digital Riken atlas¹⁰⁷ from BSINI Marmoset (<http://brainatlas.brain.riken.jp/marmoset/>).

8. The existence of activation foci in subcortical structures is mentioned in the Results, but not discussed. Have corresponding AON regions been also described in macaques and humans? What is known about the possible function of the anteromedial thalamus in the context of action observation?

We appreciate the reviewer drawing our attention to this matter. We have now incorporated this result into the Discussion section. To our knowledge, only a few recent studies have identified subcortical responses in humans during action observation (Chen et al., 2023; Errante & Fogassi, 2020; Palermo et al., 2020). Specifically, Errante and Fogassi compared the observation and execution of meaningful actions and simple finger tapping, observing the activation of the pulvinar and the subthalamic nucleus, among other subcortical activations detected. However, these authors did not report activation of the anterior portion of the thalamus, which we observed in our sample. Consequently, it is challenging to determine the role this structure may play, given the limited available data. We suggest that further studies exploring a greater variety of experimental contexts and/or investigated actions are crucial for better understanding the thalamic involvement in the action observation circuit. The main text has now been changed accordingly:

Page 14, line 302:

In conclusion, our findings further support the existing literature on the human AON, specifically regarding subcortical activation at the thalamic level. Thalamic activation during the observation of goal-directed actions has been observed in both healthy humans^{87,88} and patients with Parkinson's disease⁸⁹. Errante and Fogassi, for instance, examined subcortical responses during the presentation and execution of meaningful actions and simple finger tapping actions⁸⁸. The pulvinar activation they identified aligns with the subcortical activations reported in our study. Additionally, the marmoset AON encompasses a large bilateral portion of the anteroventral and anteromedial thalamus. To our knowledge, the

involvement of this region has not been reported in the literature on the AON for either humans or macaques. Therefore, further studies using different types of actions or contexts are desirable to better understand this finding.

Reviewer #2 (Remarks to the Author):

This is a first study of brain activations in the marmoset in response to observation of goal-oriented reach behavior, as previously studied in humans and macaque monkeys. Marmosets are provided videos to observe in a 9.4T MRI. Videos consist of 'goal-directed (Grasping hand)' and 'non goal-directed (Empty hand)' grasp. Comparison of the fMRI images of goal-directed vs non goal-directed revealed primary activations in the prefrontal (areas 6, 8, 47, 11, 13), dorsomedial prehension circuits (areas MIP, LIP, PG, V3A, V6), some regions in V1, V2, V4, and temporo-occipital (areas V4, V4T, V5, TEO, FST, PGa-IPa, TE3), retrosplenial (30), and thalamic (pulvinar, LD, anteroventral) areas. These areas are largely similar to previously described action observation networks in macaques and humans. This study shows that it is possible to study behaviorally-related brain activations in marmosets in ultrahigh field MRI, further helping to establish marmosets as a model for brain studies. Some concerns are listed below.

We thank the reviewer for the careful analysis of our study and for the time invested in providing feedback. To respond to the different comments, we have changed the main text and reported the changes here. Below are the point-by-point answers.

Major comments:

1. The interpretation that the Empty hand is non goal-oriented needs some more support. Was there a different target or an internally generated goal? Was there a goal but the target was missing? When does an animal extend the arm/hand without purpose?

Thank you for drawing our attention to this. The lack of a sample of the videos we used (highlighted also in your major point #4) probably contributes to generating doubts about our control condition. The Empty Hand videos were recorded in the same conditions as the Grasping Hand videos: a marshmallow was placed in front of a small opening on the transportation box of the "actors" marmosets. Through this opening it was possible for the animal to extend an arm, reach the marshmallow and grab it. The difference between the two conditions lies in the presence of the target in the videos used. In the Grasping Hand condition, nothing has been removed, and the successful reaching-to-grasp movement can be observed. In the Empty Hand condition, however, the reaching and grasping movements are present, but the target has been removed from the video. Consequently, the observed movement, initially purposeful, loses its original intention in the final form of the video. We added in the main text a link to our videos, and we modified the Methods part trying to explain better the Empty Hand condition (page 18, lines 405-408):

“This approach enabled us to achieve a complete reaching-to-grasp movement that appeared goalless (as the target was not visible in the final video) while circumventing the challenge of training our marmosets to perform an arm extension without a purpose.”

2. The conclusion that there is no difference between eye movements during the

goal-directed vs non goal-directed videos needs additional analysis. Eye movement traces need to be further quantified as current analyses are a bit simplistic. Saccade frequency and amplitude are important aspects of eye movements but do not characterize the locations or distributions of eye movements, something which is relevant to the importance of the goal in each condition. Neither does it monitor internal tracking of a goal prior to or expected behaviors after acquisition of the goal. For example, it might be expected that during goal oriented behavior, more eye movements towards the goal or prior to achieving the goal would occur, or there might be covert attention towards the location of the goal.

We thank the reviewer for this interesting suggestion. We took this opportunity to conduct some additional analysis on the data of our eye-tracking experiment. We first divided each video into four different epochs: pre-hand (before the appearance of the limb, with target already present in the Grasping Hand condition), reaching (arm appears and complete the extension movement), withdrawing (arm has reached maximum extension and performs the bending movement) and post-hand (arm is no longer present). In the Grasping Hand condition, the target was the marshmallow, already present from the pre-hand stage. For the Empty hand condition, where there is no physical target, we detected the position reached by the hand when the arm is at its maximum extension, and we marked that position as "target". We then drew a window of 3 visual degrees around the target position and a second window centered on the hand. In this way we could calculate the time spent by each animal looking at the window centered on the target and the one centered on the hand, and we then compared these proportions of time separately for the two visual windows using two repeated-measures ANOVA with Condition (Grasping vs Empty Hand) and Epochs (pre-hand, reaching, withdrawing and post-hand) as factors. In this way, we could observe that there are some significant differences between the two conditions, for both visual windows. As for the window centered on the hand, it is observed that marmosets observe longer the limb in the phase of withdrawing in the Grasping Hand condition, once the marshmallow have been acquired. In the analysis focused on the target, on the other hand, it is observed that in the goal-directed condition the marmosets fix longer the target during the reaching phase, but not in other epochs, and in particular **not** in the pre-hand phase. Overall, these results seem to agree with what the reviewer suggested and what we hypothesized: the presence of marshmallow, per se, does not induce differences in basic saccade metrics. However, this target becomes a relevant element when the arm appears on the screen: it's at that point that the action is perceived as a goal-directed, as the presence of a target and an arm in extension towards it conveys a purpose. After acquiring the target, the eye focuses more on the hand, following it in its trajectory, but only within the goal-directed action.

These findings suggest that while the overall amplitude and frequency of the saccades does not differ in the two experimental conditions, phase-specific differences emerge in eye movements directed both toward the arm and toward the target. However, the fact that these differences, and specifically the longest observation of the target, do not emerge before the appearance of the limb or after its disappearance pushes us to argue that the brain activations found in our fMRI experiment are not explainable only as a function of the different oculomotor pattern. Instead, it seems more correct to assume that the discrimination between the goal-directed and the non-goal-directed action takes place just when the target and the limb are associated.

We have therefore modified the main text in more points to insert this new analysis, its methodology and its results. For example, in the results section, on pages 9 and 10 it now appears:

Moreover, considering the presence of a target in the Grasping Hand videos and its absence in the Empty Hand condition, we investigated separately the eye movements towards the limb and target positions throughout the duration of the videos (the "target" position in the Empty Hand condition was considered as the position of the maximum arm extension achieved in each video, see more in details in Methods section). According to our hypothesis, the presence of a target (i.e., the marshmallow) becomes relevant when the arm appears on the screen, because it is the association of the limb and the target to make the perceived action "goal-directed". We thus divided each video into four Epochs: pre-hand (before the appearance of the limb, with target already present in the Grasping Hand condition), reaching (arm appears and complete the extension movement), withdrawing (arm has reached maximum extension and performs the bending movement) and post-hand (arm is no longer present). Two repeated-measures ANOVA were conducted separately to analyze the time spent looking within a 3 visual degree radius window centered on the hand or target, with the factors of Condition (2 levels: Grasping Hand and Empty Hand) and Epoch (4 levels: pre-hand, reaching, withdrawing and post-hand). As for the time spent looking at the hand, it was observed that both main effects were significant (Condition: $F_{(1,7)}=5.96$, $p=0.045$, Epoch: $F_{(2,3,16.1)}=44.68$, $p<0.001$), as well as their interaction (Condition*Epoch: $F_{(2,1,14.7)}=3.73$, $p=0.047$ and $\eta^2_p=0.347$). Post-hoc comparisons with Bonferroni's correction revealed a significant difference between the Grasping (mean=0.179, SE=0.019) and Empty Hand (mean=0.13, SE=0.006, with $p=0.02$) conditions once the hand has reached the target and retracts (withdraw epoch), with more time spent watching the hand in the case of goal-directed action. On the other hand, the ANOVA conducted on the time spent observing the target revealed a significant effect of Epoch ($F_{(1.74, 15.7)}=27.15$, $p<0.001$) and of its interaction with Condition (Condition*Epoch: $F_{(1.4, 12.64)}=4.4$ $p=0.046$ and $\eta^2_p=0.328$). Post-hoc comparisons again revealed a significant difference between the two conditions, this time inherent the reaching phase, with greater time spent looking at the target during the goal-directed action (mean=0.135, SE=0.025) than during the non-goal-directed condition (mean=0.074, SE=0.011 with $p=0.048$).

Altogether, these results (reported in Figure 3) suggest basic saccade metrics are similar between the two experimental conditions, indicated by the absence of differences in the frequency and amplitude of the saccades. However, the gaze patterns do differ slightly. More specifically, the target and the limb attract gaze when on the screen simultaneously as part of a goal directed action. This does not happen in the case of a non-goal-directed action.

Figure 3. Frequency (A panel, number of eye movements per second with a radial eye velocity greater than 30 deg/s and amplitude greater than 0.35 visual degrees) and amplitude (B panel, in visual degrees) of saccades performed during the presentation of videos depicting goal-directed (Grasping Hand condition, in red) and non-goal-directed actions (Empty Hand condition, in orange). No differences between conditions emerged using Wilcoxon signed rank test (in both cases, $p > 0.6$). Error bars represent the standard error. Panels C and D represent the proportion of time spent observing a window of 3 visual degrees centered on the hand or target, respectively. This proportion is reported for the two experimental conditions (Empty Hand, in orange, and Grasping Hand, in red) across the four epochs (pre-hand, reaching, withdrawing and post-hand). Post-hoc comparisons with Bonferroni correction showed a significant difference between the two experimental conditions during the withdrawing phase regarding the visual window centered on the hand (C panel) and during reaching for the visual window centered on the target (panel D). Asterisks indicate such significant differences: $*p < 0.05$. Black lines and squares represent median and mean of the group respectively; black dots represent individual scores.

In the Discussion section, from page 12 line 323 to page 16 line 346, we now added:

An interesting result comes from the epoch-by-epoch eye-tracking analysis, in which a significant difference is observed between the oculomotor patterns in the goal-directed and non-goal-directed conditions. In particular, our marmosets seem to focus more on the

marshmallow in the Grasping Hand condition, compared to the "target" position of the Empty Hand one, although these two targets are largely spatially overlapped on the screen. This difference, however, only emerges during the reaching phase, when the arm appears and performs the extension movement. It is important to note that the target does not elicit any difference in the oculomotor behavior before that phase: when only a marshmallow (Grasping Hand, pre-hand phase) is represented on the screen or no target is presented (Empty Hand, pre-hand phase), the two experimental conditions do not differ. This result supports our fMRI findings, as it seems unlikely that the mere presence of a reward (i.e., the marshmallow), without inducing a specific oculomotor behavior, is capable of inducing the brain activations we reported. It is more likely that the attribution of a purpose to the action is due to the association of the limb and the target: while the time spent watching the hand during the reaching phase in the Grasping Hand and Empty Hand conditions does not differ, the greater time spent looking at the target in the Grasping Hand videos can explain the attribution of a purpose to such movement. This interpretation is in accordance with what previously observed in macaques⁶⁵: the perception of the initial phase of a goal-directed movement is already able to activate the mirror neurons system, even when the final part of the action is obscured, but this happens only if, before the action, a target is presented. If, in the opposite case, the object is not shown (as in our Empty Hand condition), the mirror neuron system remains silent. Altogether, the results of the eye-tracking study therefore seem to provide additional support to our fMRI results, confirming in the marmosets the presence of an action-observation network similar and comparable to that described in both humans and Old-World macaques.

Finally, in the Methods section, from page 22 line 538 to page 23 line 556, we modified accordingly:

Furthermore, each video was divided into 4 epochs, based on the content: pre-hand (arm not yet present), reaching (arm appears and performs the reaching movement up to its maximum extension), withdrawing (arm has reached its maximum extension and performs the bending movement until it disappears from the screen) and post-hand (arm no longer present). In each of these phases, a window of 3 visual degrees was drawn around the hand, following its trajectory (in the pre-hand and post-hand phases the window includes the portion to the far right or left of the screen, depending on the type of video - leftward or rightward reaching). A second window of 3 visual degrees was also drawn around the target: the marshmallow in the Grasping Hand condition and the position reached by the hand following the maximum extension of the arm in the Empty Hand condition. This last window, like the previous one, followed the position of the target during the whole video (in the Empty Hand condition, the target window coincided with the hand window starting from the withdrawing phase). In this way, it was possible to compute the proportion of time spent watching the hand or target during the four epochs. The differences between the different epochs and/or the two experimental conditions were then analyzed through two distinct repeated-measures ANOVA (one for the visual window centered on the hand, one for the centered on the target) with factors Condition (Grasping Hand vs Empty Hand) and Epochs (pre-hand, reaching, withdrawing, post-hand). Where necessary, violations of sphericity were corrected by Greenhouse and Geisser correction. Any significant effects were further investigated through post-hoc comparisons with Bonferroni correction.

3. Are there any goal-directed behaviors that are unique to marmosets? This might add to novelty of the study.

We thank the reviewer again for the interesting insight. This aspect was an important point of reflection for us in designing our study. However, to our knowledge, there is no such "unique" behavior for marmosets, and in particular a type of behavior that can be tested in an fMRI setting. We therefore preferred to opt for a behavior previously studied and tested on macaques (as in Nelissen et al., 2005, Science). Anyway, our idea is that it is possible to test different types of behavior, more or less suitable for the specific situation: as demonstrated by Ngo et al., 2022 (Current Biology), marmosets tend to capture prey with reduced avoidance capabilities (such as ants or termites) using the mouth, rather than the hand. The marshmallow in our studio would therefore fall into this category of "prey", but the setting we used in our videos prevents this type of approach. We are therefore currently testing the responses of this action-observation network in front of two different types of goal-directed action: grasping with the mouth versus grasping with the hand.

4. Example of one set of 8 videos should be provided

Thank you, we now added an example of all the videos we used in our study: Grasping Hand, Empty Hand, Scrambled Grasping Hand and Grasping Empty Hand (both in their original and mirrored form). These videos are available on OSF, at the same link we reported in the text in the [Data and Code Availability Statement](https://osf.io/hvbmj/?view_only=6ea57106e8ce464fb0574a1acbc2f89d) (https://osf.io/hvbmj/?view_only=6ea57106e8ce464fb0574a1acbc2f89d)

Other comments

1. Some basic things are missing. I would like to see some BOLD signal timecourses, single condition activation maps, and eye movement traces.

We now modified Figure 2 to report an example of BOLD signal timecourse for our experimental conditions and we added two supplementary figures showing single condition activation maps and an example of eye movement traces for each type of video we used in our eye-tracking experiment (Grasping Hand and Empty Hand, both from the left of the right side).

We modified the main text accordingly:

Page 4, lines 93-94:

See Supplementary Information for individual maps and for single condition activation maps.

Page 6: Figure 1 has been modified to include BOLD timecourse of one run for two key regions of the AON: area 45 and FST

Page 8, lines 144-146:

An example of eye movement traces for each type of video is available in Supplementary Information (Figure S5).

Single condition activation maps

Figure S4. Functional maps of the contrasts between each condition and baseline. Grasping Hand (A), Empty Hand (B), Scrambled Grasping Hand (C) and Scrambled Empty Hand (D) maps are displayed on the left and right fiducial brain surfaces (lateral view on the left, medial view on the right). White lines delineate the cerebral areas included in the Paxinos parcellation of the NIH marmoset brain atlas (Liu et al., 2018). The responses here reported have intensity higher than $z = 3.29$ (corresponding to $p < 0.001$, AFNI's 3dttest++) and survived the cluster-size correction (10000 Monte-Carlo simulations, with $\alpha = 0.05$).

Supplementary information, page 44:

Figure S5. Eye movement traces for Grasping and Empty Hand videos used in the eye-tracking experiment, for both actions performed from the left and from the right side of the screen. The black and red lines represent the horizontal and vertical components of eye movements respectively. For each type of video, the dashed lines represent three key moments: the beginning of the movement (the arm appears on the screen), the reaching of the goal (in Grasping Hand, grasping of the marshmallow, in Empty Hand, reaching the point of maximal extension of the arm for that specific video) and the end of the action (arm disappears from the screen).

2. Please clarify: the 8 videos are Grasping Hand, Empty hand, Scrambled Grasping Hand, Scrambled Empty Hand, and the mirror reversals of these videos?

Indeed, we initially recorded 12 distinct videos for both the Grasping Hand and Empty Hand conditions, featuring an arm performing a left-to-right movement. We then mirrored these videos, yielding 24 videos per condition (half left-to-right and half right-to-left movements). Lastly, we phase-scrambled them, creating 24 videos for both the Scrambled Grasping Hand and Scrambled Empty Hand conditions. However, we never treated leftward and rightward videos as separate conditions. Thus, we have four experimental conditions (Grasping Hand, Empty Hand, Scrambled Grasping Hand, and Scrambled Empty Hand), encompassing movements performed in two different directions.

In each run, we included eight stimulation blocks: two per condition, with rightward and leftward movements intermingled. We have revised several lines in the Methods section (page 19) to better clarify this aspect.

3. Are there differences in pupillary dilation in goal-oriented vs non goal-oriented tasks?

We thank the reviewer for the suggestion. Unfortunately, in our eye-tracking study, we recorded data on eye movements, but we don't have data on pupillary dilation. The main reasons why we do not record these data are the large pupillary size of marmosets associated with the tendency to slightly close the eyelids. In these circumstances, the pupillary dilation signal is often lost, making the data unreliable or insufficient for analysis. We therefore decided not to record this data and to focus our attention on eye movements.

4. Where is the target in visual field?

We thank the reviewer for raising this important point. We noted that we had not previously entered any information regarding the size of the visual window occupied by the videos or the position of the action's target.

In the different Grasping Hand videos, the position of the target was slightly varied, but consistently within the central visual field: the average distance of the center of the marshmallow from the center of the screen is 3.3 cm, equivalent to about 1.6 visual degrees (ranging from 0 to ~3 visual degrees). These dimensions have been matched in the eye-tracking experiment, when scaling the videos.

We have changed the text accordingly (page 18-19, lines 432-435):

The visual window occupied by the videos, centered on the screen, is a 25*14 cm (12°*6.7°) rectangle. The target (when present) fell always within the central visual field (min/max distance of the center of the marshmallow from the center of the screen: 0°/~3°).

5. UHF causes nystagmus? Does this cause discomfort to the animal?

We subjectively observed that, as for humans, the ultra-high magnetic field may induce nystagmus in some animals. This can be more pronounced in some animals and completely absent in others. Of the 7 animals tested, only one exhibited a strong nystagmus at each MRI session, but this nystagmus was quickly reduced within minutes when we presented a fixation point on the screen. Such a reference can provide an "anchor" for the animal's gaze, as previously observed even in humans, and thus reduce the possible discomfort caused by the magnetic field.

None of the animals presented nystagmus during the presentation of the test blocks.

We added in the text (page 19, line 438) the reference for the study on human participants describing the effect of the central fixation point on the vestibulo-ocular reflex (Gauthier & Vercher, 1990):

"During the baseline, a black filled circular shape (1.5 cm of diameter, 0.72°) was presented at the center of the screen, but the animals were not required (nor trained) to fix: the only purpose of the black dot was to reduce the nystagmus sometimes present because of the ultra-high magnetic field, as previously observed in humans⁸⁹.

6. Please provide a reference to the behavior 'grasp with mouth'

Thank you for drawing our attention on this. We modified the main text adding the reference Ngo et al., 2022 (page 14, lines 288-289):

"marmosets often use their mouth to capture "preys" with limited avoidance behaviors, if left free to choose⁸⁰"

Reviewer #3 (Remarks to the Author):

In this study, 7 awake common marmosets (*Callithrix jacchus*) were presented with videos showing the arm and hand of another marmoset making movements to either a visible goal (grasping a piece of marshmallow) or an invisible goal (same videos with the marshmallow deleted making the hand look as if it was grasping air). Brain responses to these experimental conditions were examined at a magnetic field strength of 9.4 T using an fMRI block-design alternating the actual videos and their scrambled versions for ten runs per animal, that is, a total of 70 runs. This was completed by an out-of-the-scanner eye-tracking experiment in 10 monkeys confirming that Reaching & Grasping Food and Reaching & Grasping Air videos elicited similar oculomotor patterns.

New World marmosets have long attracted neuroscientists' interest. Prosocial, they might more closely approximate human cooperative social behavior than the competitive Old World macaques having heretofore dominated nonhuman primate neuroscience. This alternative primate model will likely gain even further importance in the future given the dire shortage of lab macaques that plagues worldwide research in the wake of Covid19 pandemic.

High-field fMRI is a state-of-the-art technique mastered by few neuroscientists. The methods used to collect and analyze imaging data should thus be evaluated by an expert. I am not competent to critically evaluate the authors' choices. Otherwise the paper is well-written easy to read, the behavioral methods are sound and the imaging results are clearly described and illustrated. I have but only minor comments and questions.

We would first like to thank the reviewer for the careful analysis of our manuscript and the methodology we used. The comments provided allowed us to ameliorate and clarify several aspects of the main text. We have modified the manuscript to respond as accurately as possible to the points raised. We report below the answers point-by-point.

1. Could the authors provide information about the monkeys' prior experience? Were they naïve or on the contrary highly familiar with fMRI scanning?

We thank the reviewer for highlighting this aspect. All seven animals in our sample had prior fMRI scanning experience. The least experienced monkey had already completed at least four other fMRI sessions before our current experiment, while the most "experienced" one had completed 30. In addition to their firsthand experience, all the animals underwent a three-week training period before the first actual fMRI session, during which they were gradually acclimatized to the fMRI scanning environment. We have modified the manuscript accordingly (page 16, lines 368-370):

"All animals included in the sample had previous experience with the fMRI scanning setting, ranging from 4 to 30 fMRI sessions performed prior to this experiment"

2. Monkeys received no reward to look at the videos during imaging and eye tracking. Does it mean that, unlike macaques, marmosets do not need any water or food

control to comply with testing? Did they steadily look at stimuli while in the scanner?
How many runs had to be discarded out the 70 acquired runs?

Regrettably, despite their recent success and the strong interest of the neuroscientific community, marmosets, like macaques, require training if test conditions demand a fixation task or prolonged visual exploration. In our study, where there is no reward and the animals are not under food or water restriction, compliance relies primarily on the individual animal. We employed short runs (~5 minutes) to minimize the duration of the task, which helps maintain the animal's attention in most cases. However, while some animals can fixate on the screen for 4-5 runs per session, with other marmosets, it is often only possible to obtain 1 or, at most, 2 usable runs per session. After that time, the animal tends to close its eyes and ignore the videos.

In our study, we used a run selection criterion based on online qualitative analysis of the animal's state (described on page 19, lines 436-439). The MRI-compatible camera we used allowed us to monitor the monkey's wakefulness during each run. Runs in which marmosets closed their eyes for more than one entire stimulation block, regardless of the experimental condition, were excluded. In total, 32 runs were excluded from analysis (before preprocessing or any other form of data analysis). This information has now been incorporated into the manuscript, on page 19, line 439:

The compliance of the animal during each run was checked and noted online by the investigator; runs in which the animal closed its eyes for two or more stimulation blocks (regardless of the experimental condition) were discarded from further analyses (N = 32).

To address the reviewer's interest further: in recent experiments utilizing visual stimuli, we have begun to implement a reward system linked to the animal's eye-opening. This reward is provided as long as the animal observes the screen but does not require fixation on a specific point. This approach significantly reduces the number of sessions needed to complete a study and eliminates the need for training. However, it also introduces a potential confounding factor related to the reward aspect.

3. Rather than "Old World primates" I would use "Old World macaques" to be more accurate.

We thank the reviewer for the clarification: we modified the paper to apply this correction.

4. I may have missed it but I could not find a link to samples of the videos.

Thank you again, we now added an example of all the videos we used in our study: Grasping Hand, Empty Hand, Scrambled Grasping Hand and Grasping Empty Hand (both in their original and mirrored form). These videos are available on OSF, at the same link we reported in the text in the Data and Code Availability Statement (https://osf.io/hvbmj/?view_only=6ea57106e8ce464fb0574a1acbc2f89d)

Reviewers' comments:

Reviewer #1 (Remarks to the Author):

The authors have addressed all my comments on the previous version, and I have no further points to raise.

Reviewer #3 (Remarks to the Author):

The authors fully addressed my concerns in their response, and made extensive revisions to comply with the requests of the other reviewers.

So I think the paper should now be accepted for publication.

Reviewer #4 (Remarks to the Author):

Marmoset is a highly vocal and highly social New World monkey, which has attracted much attention from neuroscientists. It is an alternative non-human primate model for study of social, prosocial, and human cooperative behavior. In this study, ultra-high field fMRI at 9.4T was performed when awake common marmosets were watching videos depicting goal-directed (grasping food) or non-goal-directed actions. The observation of others' actions activates a network of temporal, parietal, and premotor/prefrontal areas in marmosets, which is similar to macaque monkeys and humans. This is a very interesting study. Most of the reviewers' comments have been addressed in the revision, and the manuscript has been greatly improved. I have a few minor comments on the manuscript.

1. Figure S1 shows individual maps and their overlapping for intact vs scramble movement comparison. The individual variation is much larger than expected. Additionally, which individual is a left-handed marmoset? The authors proposed two possible reasons for the laterality of the response (figure 1). Does the result of this monkey match the first reason that the authors raised? I am curious about this result. Personally, I don't think the laterality of the response (figure 1) is due to left- or right-handedness.

2. Page 3, line 63, please cite the references for wireless and datalogger recording techniques;

3. Page 3, line 65, please cite the references.

In reviewing the manuscript, please consider the following questions:

Does the manuscript have technical or conceptual flaws that should prohibit its publication? If so, please provide details. In Part

Are the conclusions original? If not, please provide relevant references. Yes

Do you feel that the results presented are of immediate relevance for people in your own discipline or for a broader audience? Yes

If you recommend publication, please outline briefly what you consider to be the outstanding features. This study is very interesting but the interpretation of the results is not convincing.

Response to reviewers

Reviewer #4 (Remarks to the Author):

Marmoset is a highly vocal and highly social New World monkey, which has attracted much attention from neuroscientists. It is an alternative non-human primate model for study of social, prosocial, and human cooperative behavior. In this study, ultra-high field fMRI at 9.4T was performed when awake common marmosets were watching videos depicting goal-directed (grasping food) or non-goal-directed actions. The observation of others' actions activates a network of temporal, parietal, and premotor/prefrontal areas in marmosets, which is similar to macaque monkeys and humans. This is a very interesting study. Most of the reviewers' comments have been addressed in the revision, and the manuscript has been greatly improved.

We thank the reviewer for the careful analysis of our manuscript and our responses to previous revisions. We believe that our responses and the changes we made to the text as a result of her/his suggestions are able to increase the quality of our manuscript.

I have a few minor comments on the manuscript.

1. Figure S1 shows individual maps and their overlapping for intact vs scramble movement comparison. The individual variation is much larger than expected. Additionally, which individual is a left-handed marmoset? The authors proposed two possible reasons for the laterality of the response (figure 1). Does the result of this monkey match the first reason that the authors raised? I am curious about this result. Personally, I don't think the laterality of the response (figure 1) is due to left- or right-handedness.

We thank the reviewer for raising this point and we agree with her/him about the inter-individual differences within our sample. Such differences, in our experience, are not a rarity: in multiple works using fMRI in awake marmosets from our laboratory, individual results often show this type of variability. This variability does not seem to be linked specifically to the observation of goal-directed actions, as we found it also during the visual perception of social or non social scenes (Clery et al., 2021) or while listening to marmoset's vocalizations (Jafari et al., 2023). We don't have, at the moment, a sure answer to such variability, but several factors can concur: the tSNR of our acquisitions is not always constant, and varies in function of the marmoset and the position of the headpost (slight variations in headpost inclination can cause the position of the head to vary from the two parts of our coil, thus changing the quality of the signal locally). Another factor may be the compliance of the animal: for some marmosets it is quite easy to get functional runs where the animal pays attention and actively explores the screen; for others, it is necessary to perform multiple functional runs in order to get some with good compliance. However, repeating the same videos may induce habituation in marmoset, and this may then cause differences between the activations of different animals. Finally, very often studies using fMRI on humans report only the second level analyses, not showing the activations at the individual level. It is not excluded that this type of variability may also be present in this type of study. However, the representation of the activation frequency of each voxel (Figure S1 and S2, top panel) seems to show that within our sample there is a good overlapping of brain responses in front of the presentation of our videos, with peaks of 5 monkeys (thus more than 70% of the sample) presenting activations in the superior temporal areas included in the AON, for example.

As for the handedness of our sample, the only left-handed animal is M3 (in figure S1 below referred to as "Left-Handed"). As we propose in the manuscript (Supplementary Materials, page 38, from line 879, here reported highlighted in yellow), the presence of a single left-handed animal does not allow us to go beyond mere speculation. Also, as stated in the text below, the activation pattern of our animal alone is unable to confirm this speculation. Such effect of lateralization of the activations, therefore, remains for now not explainable in function of the handedness, but such aspect is sure a point to investigate deeply in future studies.

An interesting aspect, but that we cannot investigate in depth because of the composition of our sample, concerns the manual preference shown by our marmosets. As already amply demonstrated in literature¹¹²⁻¹¹⁴, marmosets often show a strong manual preference in tasks of reaching for food, and such preference may reflect hemispheric dominance of other cognitive domains¹¹⁵. Unfortunately, only one (M3) of the 7 animals we tested shows a preference for the use of the left hand, and we are therefore unable to investigate statistically differences in the AON related to this preference. In addition, the lack of separation of videos representing actions carried out with the left hand and with the right hand in our experimental design adds a degree of uncertainty to any possible interpretation. On a purely qualitative level, the Figures S1-C and S2-C do not seem to report a different pattern of activations for M3 compared to the 6 right-handed animals.

The impossibility to test the relationship between the manual preference of marmosets and the extension/lateralization of their action observation network is therefore a limit of this study and a possible starting point for future work.

We also added a more specific sentence regarding the absence of (qualitative) differences between the individual activations of M3 (the left-handed marmoset) and the rest of the sample (page 12, line 225):

Moreover, the pattern of activations of our left-handed animal does not differ, from a qualitatively point of view, from the rest of the sample.

2. Page 3, line 63, please cite the references for wireless and datalogger recording techniques;
3. Page 3, line 65, please cite the references.

We thank the reviewer for pointing this out. References for wireless recording and marmosets optical imaging have now been added to the main text (references 40, 41 and 42) in page 63.

REVIEWERS' COMMENTS:

Reviewer #4 (Remarks to the Author):

The authors have addressed all my comments on the previous version, and I have no further points to raise.